# *In-silico* approaches for identification of compounds inhibiting SARS-CoV-2 3CL protease

**Md. Zeyaullah[1], Nida Khan[2], Khursheed Muzammil[3], Abdullah M. AlShahrani[1], Mohammad Suhail Khan[3], Md. Shane Alam[4], Razi Ahmad[5]\*, Wajihul Hasan Khan[6]\***

**1** Department of Basic Medical Science, College of Applied Medical Sciences, Khamis Mushayt Campus, King Khalid University (KKU), Abha, Kingdom of Saudi Arabia (KSA), **2** Department of Chemical Engineering, Indian Institute of Technology Delhi, New Delhi, India, **3** Department of Public Health, College of Applied Medical Sciences, Khamis Mushayt Campus, King Khalid University (KKU), Abha, Kingdom of Saudi Arabia (KSA), **4** Department of Medical Laboratory Technology, College of Applied Medical Sciences, Jazan University, Jazan, Saudi Arabia, **5** Department of Chemistry, Indian Institute of Technology Delhi, New Delhi, India, **6** Department of Microbiology, All India Institute of Medical Sciences Delhi, New Delhi, India

\* razi.jmi@gmail.com (RA); wajihulbiotech@gmail.com (WHK)

**Data Availability Statement:** All relevant data are within the manuscript and its Supporting Information files.

## Abstract

The world has witnessed of many pandemic waves of SARS-CoV-2. However, the incidence of SARS-CoV-2 infection has now declined but the novel variant and responsible cases has been observed globally. Most of the world population has received the vaccinations, but the immune response against COVID-19 is not long-lasting, which may cause new outbreaks. A highly efficient pharmaceutical molecule is desperately needed in these circumstances. In the present study, a potent natural compound that could inhibit the 3CL protease protein of SARS-CoV-2 was found with computationally intensive search. This research approach is based on physics-based principles and a machine-learning approach. Deep learning design was applied to the library of natural compounds to rank the potential candidates. This procedure screened 32,484 compounds, and the top five hits based on estimated $pIC_{50}$ were selected for molecular docking and modeling. This work identified two hit compounds, CMP4 and CMP2, which exhibited strong interaction with the 3CL protease using molecular docking and simulation. These two compounds demonstrated potential interaction with the catalytic residues His41 and Cys154 of the 3CL protease. Their calculated binding free energies to MMGBSA were compared to those of the native 3CL protease inhibitor. Using steered molecular dynamics, the dissociation strength of these complexes was sequentially determined. In conclusion, CMP4 demonstrated strong comparative performance with native inhibitors and was identified as a promising hit candidate. This compound can be applied in-vitro experiment for the validation of its inhibitory activity. Additionally, these methods can be used to identify new binding sites on the enzyme and to design new compounds that target these sites.

**Funding:** This research was funded by the Deanship of Scientific Research at King Khalid University, Abha, KSA through a research group program under grant number RGP. 2/181/43. The funders had no role in study design, data collection and analysis, decision to publish, or preparation of the manuscript.

**Competing interests:** The authors have declared that no competing interests exist.

## 1. Introduction

The first case of atypical pneumonia, severe acute respiratory syndrome (SARS), was observed in China Guangdong province, and since then it has spread to several other countries. Coughing, a high temperature, chills, convulsions, headaches, dizziness, increasing radiographic abnormalities of the chest, and lymphopenia are the most typical SARS symptoms. In recent times, this viral infection has been renewed into the most lethal coronavirus pandemic in 2019 caused by the SARS-CoV-2 (Severe Acute Respiratory Syndrome Coronavirus-2), which spread globally and resulted in significant fatalities [1, 2]. According to the genomic data analysis, SARS-CoV-2 is most similar to SARS-CoV and MERS-CoV and shares 75–85% sequences similarity with other coronavirus strains [3–7]. In humans, birds, and other animals, coronaviruses have been linked to hepatitis, gastroenteritis, and other diseases [8–11]. It was found that the coronavirus infection was also sensitive to several co-morbid illnesses [9, 10, 11]. Cancer, diabetes, hypertension, cerebral infarction, chronic bronchitis, Parkinson's disease, and chronic obstructive pulmonary disease are only a few of the co-morbidities that can lead to SARS-CoV-2 infections, especially among the elderly [12–14]. The development of anti-SARS medications to prevent future outbreaks remains a serious problem on view of several variants that appeared during prolong pandemic period with compromised vaccine efficacy [2, 7, 15, 16]. This raised an alarming condition where it is critical to identify a novel drug candidate using *in-silico* based drug design approach.

There are only two known proteases encoded by the SARS-CoV-2 which include (a) papain-like cysteine protease (PLpro) [17] and (b) chymotrypsin-like cysteine protease known as 3C-like protease (3CLpro) [18–23]. The SARS-3CLpro is highly homologous with other coronavirus 3C-like proteases and is fully conserved among all the known SARS coronavirus genome sequences available. Some recent studies indicate the potential compound that could inhibit the 3Cl protease of SARS-CoV-2 [24, 25]. The Leu-Gln motif is a conserved pattern of the 3CLpro of SARS-CoV-2 and is involved at 11 positions in the cleavage of polyproteins, a process initiated by the enzyme own autolytic cleavage (autoprocessing) [19, 26]. The catalytic dyad His[41] and Cys[145] is present in the SARS-CoV2 3CLpro active site, which is positioned at the center of cleft between domains I and II [20, 27, 28]. 3CLpro is an effective therapeutic target for treating corona-viral infection since the autocleavage process is crucial in virus propagation [29]. The protease inhibitors are most effective at inhibiting replication [30–32], thus, the 3CLpro enzyme was selected as a promising target for developing effective inhibitors against SARS-CoV in this study.

The 3CLpro enzyme exists in a homodimeric state, wherein each monomer contributes to the formation of an active site. Despite this, the monomeric form of 3CLpro has also been observed to display enzymatic activity, albeit at a lower efficiency than the dimeric form. While the dimeric form is the biologically relevant form of the enzyme, analyzing the monomeric form can provide valuable insights into potential inhibitors and their mechanism of action for drug discovery. The monomeric form of 3CLpro acts as a precursor to the dimeric form, with the enzyme initially synthesized as a monomer before undergoing a conformational change that allows it to dimerize. Consequently, targeting the monomeric form may prevent the formation of the active enzyme complex. By identifying compounds that effectively inhibit the monomeric form, a better understanding of the structure-activity relationships underlying inhibitor binding can be obtained, ultimately guiding the development of more effective inhibitors. Previous studies have investigated the binding of the inhibitor N3 to the monomeric form of 3CLpro, with the PDB code 6LU7 being utilized in these in-silico analyses against SARS-CoV [33–35]. It was concluded in other studies that the binding of N3 to the dimers has an allosteric effect, which means that it allows for only one protomer at a time to be active [36].

An initial *in silico* investigation employing various computational approaches could greatly reduce the time needed for lead molecule discovery [37–39]. It is essential to determine the molecular interaction of the ligands with the target protein to estimate the therapeutic and inhibitory potential of a given compound. There has been a recent addition of a new dimension through the use of machine learning techniques with virtual drug screening methods for the creation of novel medications [33, 35, 40–46], disabling multidrug resistance [47], and applications in precision medicine to choose drugs for customised treatments [48, 49]. Several studies were reported which demonstrate the application of machine learning for predicting potential inhibitory compounds for SARS-CoV-2. In one of the study by Ton *et al.* [50] a deep docking model was applied to screen compounds from the ZINC15 library and suggested the top 1000 hits as potential SARS-CoV-2 3CLpro inhibitors. Similarly, another study used a deep learning model to predict the inhibitory activity against 3CLpro in SARS-CoV for unknown compounds in the virtual screening process, as reported by Kumari *et al.* [51]. Random forest (RF) and support vector machine (SVM) models were used in a study by Liang *et al.* to hunt novel anti-SARS-CoV-2 compounds from medicinal plants using traditional Chinese medicine (TCM) principle applying machine learning methods [44]. One study also developed a machine learning suite called "REDIAL-2020" to estimate small molecule activity from molecular structure, for a range of SARS-CoV-2 related assays [52]. Attiq *et al.* used machine learning algorithm of Flare by Cresset group which was employed with Field template, 3D-QSAR, activity Atlas model and molecular docking for FDA approved M-pro SARS-CoV-2 repurposed drugs were performed [53].

In this study, a combination of machine learning and physics-based techniques is reported to screen potential compounds against the SARS-CoV-2 3CLpro protein. Virtual screening was performed with machine learning pre-trained and deep learning models to study potential inhibitory compounds against the 3CLpro of SARS-CoV-2. The most promising compounds detected by these ML models were further used to perform molecular docking and molecular dynamics simulation to study the binding characteristics of the compounds with 3CLpro protein. Overall, this study showed the application of ML models and physics-based methods (molecular docking and MD simulation) to detect the potential compound against the 3CLpro protein and further demonstrate the detailing of the protein-ligand interaction.

## 2. Methodology

### 2.1 Machine learning

**a. Training compounds.** Pretrained models used in this study were trained on the Binding DB database [54]. Compounds that had $IC_{50}$ values reported in the database were used in training the models. The SMILES (simplified molecular-input line-entry system) of these compounds were collected and stored, which were later used to train the ML model using DeepPurpose framework. Each datapoint has a protein sequence (target) and SMILES (drug) with their corresponding $IC_{50}$. Illustrating the datapoint used these pretrained model, following example is shown:

**Drug:** CC1 = C2C = C(C = CC2 = NN1)C3 = CC (= CN = C3)OCC(CC4 = CC = CC = C4)N

**Target:** MKKFFDSRREQGGSGLGSGSSGGGGSTSGLGSGYIGRVFGIGRQQVTVDEVL AEGGFAIVFLVRTSNGMKCALKRMFVNNEHDLQVCKREIQIMRDLSGHKNIVGYIDSSI NNVSSGDVWEVLILMDFCRGGQVVNLMNQRLQTGFTENEVLQIFCDTCEAVARLHQC KTPIIHRDLKVENILLHDRGHYVLCDFGSATNKFQNPQTEGVNAVEDEIKKYTTLSYRA PEMVNLYSGKIITTKADIWALGCLLYKLCYFTLPFGESQVAICDGNFTIPDNSRYSQDMH CLIRYMLEPDPDKRPDIYQVSYFSFKLLKKECPIPNVQNSPIPAKLPEPVKASEAAAKKTQ PKARLTDPIPTTETSIAPRQRPKAGQTQPNPGILPIQPALTPRKRATVQPPPQAAGSSNQ

PGLLASVPQPKPQAPPSQPLPQTQAKQPQAPPTPQQTPSTQAQGLPAQAQATPQHQQQ
LFLKQQQQQQQPPPAQQQPAGTFYQQQQAQTQQFQAVHPATQKPAIAQFPVVSQGG
SQQQLMQNFYQQQQQQQQQQQQQQQLATALHQQQLMTQQAALQQKPTMAAGQQP
QPQPAAAPQPAPAQEPAIQAPVRQQPKVQTTPPPAVQGQKVGSLTPPSSPKTQRAGHR
RILSDVTHSAVFGVPASKSTQLLQAAAAEASLNKSKSATTTPSGSPRTSQQNVYNPSEGST
WNPFDDDNFSKLTAEELLNKDFAKLGEGKHPEKLGGSAESLIPGFQSTQGDAFATTSFSA
GTAEKRKGGQTVDSGLPLLSVSDPFIPLQVPDAPEKLIEGLKSPDTSLLLPDLLPMTDPFGS
TSDAVIEKADVAVESLIPGLEPPVPQRLPSQTESVTSNRTDSLTGEDSLLDCSLLSNPTTDL
LEEFAPTAISAPVHKAAEDSNLISGFDVPEGSDKVAEDEFDPIPVLITKNPQGGHSRNSSG
SSESSLPNLARSLLLVDQLIDL.

**Score (IC$_{50}$):** 7.365

These pretrained models were applied to an antiviral dataset to determine their applicability for ranking antiviral compounds. The compound library for testing the machine learning (ML) models was created using the ChEMBL database from EMBL-EBI [55, 56]. The virus keyword was searched in the ChEMBL database, and 500 druggable targets were found. Among them, only the single-stranded RNA viruses were chosen to filter the search hits, and 278 targets were further obtained. Afterwards, the targets were filtered with a single protein parameter, which further cut down to the list of 100 targets. Here, 32 unique protein targets (amino acid sequences) from these 100 hits were observed, these were collected and stored for feeding the data in the machine learning model as protein target sequences. Later, the IC$_{50}$ of the compounds tested against these 100 targets was searched, which resulted in 3280 compounds being obtained with their respective SMILES. Here, 2262 compounds were unique. However, the SMILES (simplified molecular-input line-entry system) of these 3280 compounds was collected and stored, which was later used to evaluate the pre-trained ML model using the DeepPurpose framework.

**b. Machine learning models.** In this study, drug screening was performed using Deep-Purpose architecture as a machine learning technique [57]. The DeepPurpose project aims to provide a simple yet powerful toolkit for drug-target interaction (DTI) prediction and its applications. It is a PyTorch-based deep learning framework that uses an encoder-decoder function to input the drug target pair and output the binding activity (here, the IC$_{50}$) of the drug target pair. Here, using the DeepPurpose framework, the ML models were used to provide the binding activity (here the IC$_{50}$) of screening compounds. The DeepPurpose Neural Network follows the first step with data feeding, where the SMILES of BindingDB compounds with their respective IC$_{50}$ values paired with the target protein amino acid sequences were fed to the model. Data encoding was performed at the encoder specification step, where the encoder was used for the SMILES of the drug and the sequence of the protein. These encoders are: (1) MPNN (message-passing neural network); (2) CNN (convolutional neural network on SMILES). (3) Morgan (Extended-Connectivity Fingerprints) (4) Daylight (daylight-type fingerprints), and (4) AAC (amino acid composition up to 3-mers). Here, Morgan and Daylight are specific for drug compounds, while AAC is used only for protein sequences. The data set during the training of the pretrained models was split into a train set, validation set, and test set with percentages of 70%, 10%, and 20%, respectively. Later, the models were configured, initialized, and trained. A neural network has multiple parameters in its training layers that were configured and optimised in these pre-trained models. Critical parameters include (a) epoch: the number of times all training datasets are iterated; (b) batch size: the number of data samples propagated through the network; and (c) learning rate: controls the size of each batch or epoch.

## 2.2 Virtual screening library

The natural compounds were screened using trained ML models on the DeepPurpose framework. Here, the PubChem database was searched to collect natural compounds where the Natural Products Atlas Classification category used and 32484 compounds were sourced [58, 59]. The Natural Products Atlas provides information on microbially-derived natural compounds and information on the source organism, which are published in the peer-reviewed primary scientific literature. Among the 32484 sourced compounds, 31401 unique compounds were observed. Later, all compounds were screened with the ML models against the 3CLpro SARS--CoV. The protein sequence 3CLpro of SARS-CoV was collected with the UniProt ID: P0DTD1 from the UniProt database [60]. These 31401 natural compounds and the amino acid sequence of 3CLpro were fed to the DeepPurpose trained ML models, and the compounds were screened based on the ranking reflected by the predicted $pIC_{50}$. Eventually, the top five compounds were selected for later use in molecular modelling analysis.

## 2.3 Molecular docking

The crystal structure of the 3CLpro in complex with the inhibitor N3 with PDB code: 6LU7 [34] was retrieved from the RCSB Protein Data Bank (RCSB PDB) database [61]. The binding pocket of the 3CLpro was determined with reference to the known inhibitor N3. PyMOL tool was used to visualize the binding site residues of the protein that covered 6 Å circular surrounding from the centre of the mass of the reference inhibitor N3 [62]. The binding pocket residues were retrieved and stored to create the grid box for the virtual screening process. This formed a grid box with dimensions of 24 Å×36 Å×30 Å on the x, y, and z axes, respectively, while it is centered at [9.07, 36.82, 79.97]. This grid box was used for the docking during virtual screening using the AutoDock Vina software [63]. The protein's 3D structure was used for docking preparation. The hydrogen atoms and charges were added to the protein molecule using the AutoDock suite and converted to a PDBQT file. The docking parameters considered during virtual screening were binding modes of 20, exhaustiveness of 100, and a maximum energy difference of 4 (kcal/mol). Initially, the top five compounds were in SMILES format, which was converted into 3D SDF files using Cactus tool [64]. Later, these 3D SDF structures were converted into PDBQT files using Openbabel tool [65]. After the docking, the best docked complex of top ligands was compared to the reference ligand and considered for intermolecular interaction analysis and molecular dynamics simulation.

## 2.4 Molecular dynamics simulation

In MD simulations, the three best hits based on the binding scores resulting from the re-docking data were selected. To comprehend the stability and flexibility of the protein-ligand complexes, MD simulation was performed for 100 ns. The chosen complexes were simulated using the GROMACS-2021 platform with the CHARMM27 force field [66, 67]. Small molecules were prepared using the CGenFF tool to generate topologies and parameters consistent with the CHARMM all-atom force field [68]. Moreover, the Ewald Particle Mesh method was used to calculate electrostatic forces [69]. The system was neutralised with $Na^+$ and $Cl^-$ ions, and the TIP3P (transferable intermolecular potential with 3 points) water model was applied to the solvation box. The complex was positioned in the middle of a solvated dodecahedron box, 1 Å distance from the wall. Later, using the steepest descent (SD) algorithm, the protein-ligand solvated complex was energetically minimized for 5000 steps. All hydrogen bonds were eliminated using the SHAKE method, and the entire system was heated to 310K [70]. The system was equilibrated to an ensemble of constant temperature (NVT) and pressure (NPT) conditions at 310 K and 1 atm, respectively, for the timeframe of 1 ns each. An equilibrated system

was used in the production run for 100 ns timescale. Temperature coupling was applied using velocity-rescaling method [71] while the pressure was maintained with the Parrinello-Rahman pressure method [72]. RMSD (root mean square deviation) and RMSF (root mean square fluctuation) were the two most important metrics used to analyse the conformation with the GROMACS internal tool.

## 2.5 MM/GBSA calculations

Using the gmx MMPBSA tool that based on the Molecular Mechanics Generalized Born Surface Area (MM-GBSA) method, the binding free energy of the protein-ligand complex was calculated [73, 74]. Last 20 ns of MD simulation trajectory, the ΔG binding free energy for the top three hits was computed. The system salt concentration was 0.154 M, and its solvation parameter (igb) was adjusted to 5. The internal dielectric constant was set to 1.0, while the exterior dielectric constant was set to 80.0. These parameters were determined based on standard values utilised in a number of comparable in-silico research [75, 76].

Here, Eq 1 shows the MM-GBSA calculation method.

$$\Delta G = < G_{complex} - [G_{receptor} + G_{ligand}] > \tag{1}$$

The $< >$ sign represents the average free energy of the complex, receptor, and ligand over the course of the last 20 ns of simulation trajectory. The equations are applied to derive the energetic components used in the ΔG computation are shown in Eqs (2–6).

$$\Delta G_{binding} = \Delta H - T\Delta S \tag{2}$$

$$\Delta H = \Delta G_{GAS} + \Delta G_{SOLV} \tag{3}$$

$$\Delta G_{GAS} = \Delta E_{EL} + \Delta E_{VDWAALS} \tag{4}$$

$$\Delta G_{SOLV} = \Delta E_{GB} + \Delta E_{SURF} \tag{5}$$

$$\Delta E_{SURF} = \gamma.SASA \tag{6}$$

Here, ΔH is the enthalpy change consisting of gas-phase energy ($G_{GAS}$) and solvation free energy ($G_{SOLV}$). TΔS represents the contribution of entropy to the free binding energy. Electrostatic and van der Waals composed $G_{GAS}$ ($E_{EL}$ and $E_{VDWAALS}$, respectively). $G_{SOLV}$ was derived from the polar solvation energy ($_{GSOLV}$) and the nonpolar solvation energy ($E_{SURF}$) was derived from the product of SASA and (solvent surface tension parameter).

## 2.6 Clustering and steered MD simulation

GROMACS g_cluster packages were used for clustering with an RMS threshold of 0.3 nm using the gromos cluster technique. The middle structure from the most populated cluster was selected for the Steered Molecular Dynamics (SMD) simulations.

In SMD simulations, a time-dependent external force is provided to the ligand to enable its dissociation from the protein, which is not possible with traditional MD simulations. In SMD, the transition between two states, bound and unbound, is achieved by adding a harmonic time-dependent potential operating on a descriptor (protein-ligand distance) with the conventional Hamiltonian. In the process of transition, the exerted force and external work produced on the system was calculated. The starting structure was collected from the clustering of the last 20 ns of the classical MD simulation that was performed earlier, and the middle structure

of the most populated cluster was used as the starting coordinate. The structure was prepared again in the SMD with the addition of charge and hydrogen atoms. The complex was placed at the centre of the cubic box of 6 Å of edge. Box was solvated with SPC water, and 100 mM NaCl salt was added. The system was energetically minimised using steepest descent minimization. NPT equilibrium was performed for 100 ps using the Berendsen pressure constant. Pulling dynamics were applied for 500 ps that used harmonic potential to pull.

## 3. Results and discussions

### 3.1 Model building and screening compounds

This study deployed pre-trained models from DeepPurpose that use the BindingDB dataset to train and test the model, this dataset consists of 2407381 datapoints with their respective protein target sequences and $IC_{50}$ values. These $IC_{50}$ values were represented in nM, and compounds were represented in their SMILES. Protein primary sequence (single letter code of amino acid) and SMILES of the compounds were encoded into a vector using different encoders. These encoders convert the amino acid sequence and SMILES into a mathematical vector that is further used as input to the machine learning model. The dataset used for training has a large range of $IC_{50}$ values to leverage diverse datapoints. Several datapoints in the dataset do not have valid numeric $IC_{50}$ values, and thus they were removed from the dataset. This reduced the dataset to 1557202 entries. The $IC_{50}$ values were converted into log scale for the more robust regression and termed as $pIC_{50}$. The maximum value of $pIC_{50}$ in the dataset was 34.53, while the minimum value was -11.51. The dataset contains 6145 unique protein sequences and 752171 unique compound SMILES. Further, the models built on the Binding DB dataset were tested on the dataset collected from PubChem on the active compounds against single stranded RNA viruses for different proteins. This data set has a $pIC_{50}$ range of 4.13 to 27.86, and the performance of the machine learning model on the known antiviral compounds could indicate the applicability of the model for the detecting the new potent antiviral compounds. There was total of 3200 data points in the known antiviral data set, where 2262 unique drugs and 32 unique protein sequences were found. Eventually, the model that outperformed the known antiviral compounds was applied as a virtual screening protocol to the natural compound library. The natural compound library contains 31401 compounds, and their respective SMILES were fed to the model along with the amino acid sequence of 3CL protease to predict the $pIC_{50}$.

   **3.1.1 ML model performance.**   Here, five different was ML models were used that trained on the Binding DB dataset. These models were different in their training parameters. Table 1 shows the parameters and their corresponding values for each model. The major difference was in the encoders uses for each models for encoding the drug SMILES and protein sequence. Here, four encoders, (1) CNN (2) Morgan (3) MPNN and (4) Daylight were used for encoding the drug SMILES. However, protein sequence was encoded using (1) CNN and (2) AAC techniques. Combination of these encoders were used to build the final predictive models.

   These models were named NET1, NET2, NET3, NET4, and NET5 as shown in Table 1. Number of train epoch that governs the exhaustiveness of the training process that reflects in the model accuracy was considered same for all the models. Thus, exhaustiveness for all the models was the same. However, the encoding methods were different, which brought variety to the model's performance. Table 2 shows the performance of each model on the external dataset that was not used in the training and testing of the model. As discussed earlier, this external dataset consists of the known antiviral compounds against ssRNA viruses for large protein targets. The performance of the model on this dataset indicates its possible applicability to antiviral compound screening.

**Table 1. Parameters used in building the predictive models trained on the Binding DB database compounds with their pIC$_{50}$.**

| Parameters | input dim drug | input dim protein | hidden dim drug | hidden dim protein | cls hidden dims | batch size | train epoch | test every X epoch | LR | drug encoding | target encoding |
|---|---|---|---|---|---|---|---|---|---|---|---|
| NET1 | 1024 | 8420 | 128 | 256 | [1024 1024 512] | 256 | 100 | 10 | 0 | CNN | CNN |
| NET2 | 1024 | 8420 | 128 | 256 | [1024 1024 512] | 256 | 100 | 10 | 0 | Morgan | CNN |
| NET3 | 1024 | 8420 | 128 | 256 | [1024 1024 512] | 256 | 100 | 10 | 0 | Morgan | AAC |
| NET4 | 1024 | 8420 | 128 | 256 | [1024 1024 512] | 256 | 100 | 10 | 0 | MPNN | CNN |
| NET5 | 2048 | 8420 | 128 | 256 | [1024 1024 512] | 256 | 100 | 10 | 0 | Daylight | AAC |

| Parameters | cnn drug filters | cnn drug kernels | cnn target filters | cnn target kernels | mpnn depth | random seed | mlp hidden dims drug | mpnn hidden size | global batch size | decay | mlp hidden dims target |
|---|---|---|---|---|---|---|---|---|---|---|---|
| NET1 | [32 64 96] | [4 8 12] | [32 64 96] | [4 8 12] | 3 | 1 | | 128 | 128 | 0 | |
| NET2 | [32 64 96] | [4 8 12] | [32 64 96] | [4 8 12] | 3 | 1 | [1024 256 64] | 128 | 128 | 0 | |
| NET3 | [32 64 96] | [4 8 12] | [32 64 96] | [4 8 12] | 3 | 1 | [1024 256 64] | 128 | 128 | 0 | [1024 256 64] |
| NET4 | [32 64 96] | [4 8 12] | [32 64 96] | [4 8 12] | 3 | 1 | | 128 | 128 | 0 | [1024 256 64] |
| NET5 | [32 64 96] | [4 8 12] | [32 64 96] | [4 8 12] | 3 | 1 | [1024 256 64] | 128 | 128 | 0 | [1024 256 64] |

*Violet colours indicate that these parameters are not applicable for those networks.

Table 2 suggest that correlation of predicted pIC$_{50}$ with the experimental pIC$_{50}$ was highest (r = 0.68) for NET1 that uses the CNN encoding for both SMILES and protein sequence. NET4 showed the minimum error value, but the correlation of the predicted pIC$_{50}$ with this model was the lowest (r = 0.06), and thus ranking the compound would not be feasible with this model. In addition, NET5 also showed a high correlation (r = 0.65) with a lower error rate compared to NET1. Daylight and AAC methods were used in the encoding of drugs and proteins during the training of the NET5 model. As per the performance shown in Table 2, both NET1 and NET5 were selected for the screening and ranking of natural compounds against 3CL protease.

## 3.2 ML screening

NET1 and NET5 were used to screen 31401 compounds using their SMILES (Simplified Molecular Input Line Entry System) and 3CL protease protein sequences. These compounds are derived from microbial cultures curated from the scientific literature and deposited at Pubchem. The Rdkit package provides a module called QED63 that stands for quantitative estimation of the drug-likeness. The QED score is based on molecular weight, logP, topological polar surface area, the number of hydrogen bond donors and acceptors, the number of aromatic

**Table 2. Performance of the Binding DB pIC$_{50}$ pre-trained models on the known set of antiviral compounds for ssRNA viruses.**

| | Correlation | Mean Absolute Error | Mean Squared error | Median Absolute Error |
|---|---|---|---|---|
| NET1 | 0.68 | 7.9 | 77.24 | 6.85 |
| NET2 | 0.64 | 8.39 | 85.58 | 7.24 |
| NET3 | 0.63 | 8.11 | 80.56 | 6.92 |
| NET4 | 0.06 | 6.8 | 68.47 | 5.9 |
| NET5 | 0.65 | 7.39 | 69.03 | 6.26 |

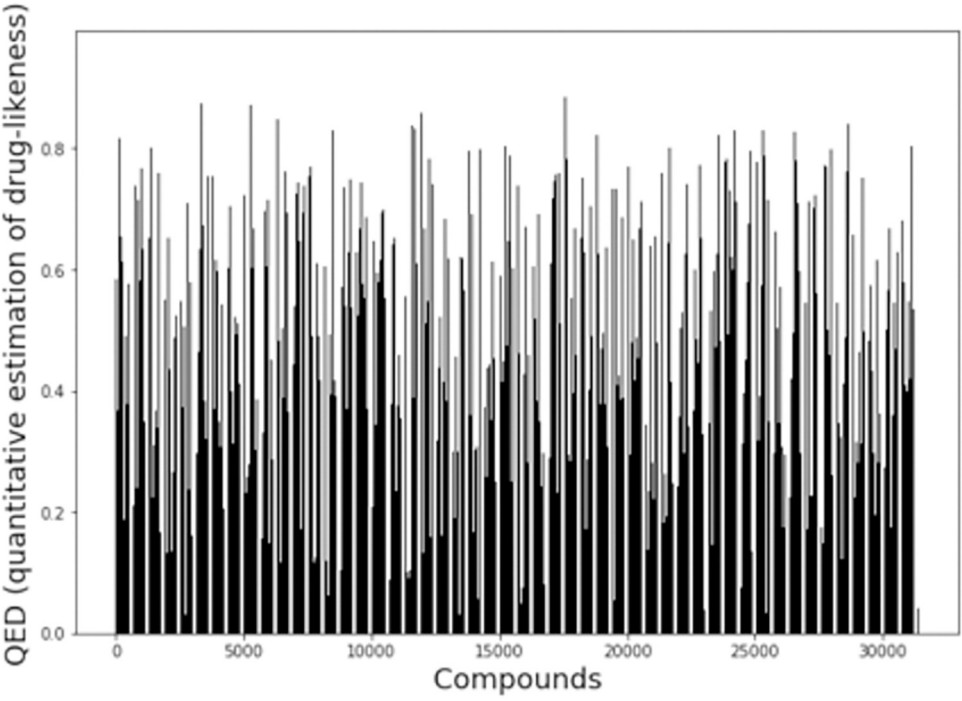

**Fig 1. Quantitative estimation of drug-likeness (QED) score of 31401 compounds from natural compound library.**

rings and rotatable bonds, and the presence of unwanted chemical functionalities for calculating the drug-likeness score. This score has a range of 0 to 1, where 0 refers to poor drug likeness and 1 signifies the maximum drug likeness. **Fig 1** shows the QED scores for all 31401 compounds from the natural compound library.

As shown in **Fig 1**, the QED scores vary for the compounds, with a minimum value of 0.06 and a maximum value of 0.94. This implies that the dataset contains compounds with low drug likeness. As the prime objective of the screening was to identify the compound with a high $pIC_{50}$ that corresponds to strong binding with the protein, a QED filter was applied post-screening to select the most drug-like candidate molecule. The NET1 and NET5 models were used on the dataset, and the top 10 unique compounds were selected based on the predicted $pIC_{50}$. High $pIC_{50}$ shows better binding and is thus preferred in this study. Table 3 shows the top 10 screened compounds from both models with their corresponding $pIC_{50}$ values.

Two dimensional representations of the top 10 molecules screened from NET1 and NET5 models are shown in the **Figs 2** and **3** respectively. Only structurally dissimilar compounds were considered in the top 10 to cover the larger sample space. $CMP4^{(NET5)}$ and $CMP10^{(NET5)}$ are very small compounds compared to other compounds. $CMP8^{(NET5)}$ does not have any ring structure, while all the other compounds screened from both models had one or more ring structures. $CMP7^{(NET5)}$ does not have any amine/hydroxyl/carboxyl group to act as donor or acceptor for forming hydrogen bonds.

Later, these compounds (NET1 and NET5) were ranked based on the QED scores, as shown in **Table 4**. Top 5 compounds were selected based on their QED scores, as highlighted 'grey' in table. Top 3 compounds in this bin were from NET5 model screening while the compounds at $4^{th}$ and $5^{th}$ positions were from the NET1 models. The QED scores for these top 5 compounds range from 0.84 to 0.56.

**Table 3. SMILES of top 10 compounds screened using NET1 and NET5 model, ranked based on their predicted pIC$_{50}$ values.**

| Name | pIC$_{50}$ | Model |
|---|---|---|
| CC(C)C[C@@H]1C (= O)N2[C@@H](C[C@]3([C@@H]2NC4 = CC = CC = C43) [C@]56C[C@H]7 <br> C (= O)N[C@@H](C (= O)N7[C@H]5NC8 = CC = CC = C68)CC(C)C)C (= O)N1 | 7.46 | NET1 |
| CC(C)[C@@H]1C (= O)N2[C@@H](C[C@]3([C@@H]2NC4 = CC = CC = C43) <br> [C@]56C[C@H]7C (= O)N[C@@H](C (= O)N7[C@H]5NC8 = CC = CC = C68)C(C)C)C (= O)N1 | 7.44 | NET1 |
| CCC (= O)[C@H](CC1 = CNC2 = CC = CC = C21)NC (= O)[C@H](CO)NC (= O) <br> [C@H] ([C@@H](C)O)NC (= O)C(C)C (= O)N[C@@H](CC3 = CC = CC = C3)C (= O)O | 7.43 | NET1 |
| CC1 = C2C[C@@](C[C@]23C (= O)[C@]4(C15CC5)CCC6 = C7 <br> [C@H]4[C@H] (O3)O[C@@]7([C@@H]8CC(C = C8C6 = O)(C)C)C)(C)CO | 7.41 | NET1 |
| CO[C@H]1C2 = NC3 = CC = CC = C3C (= O)N2[C@@H](C[C@] <br> 4([C@@H]5N1C6(CC6) (= O)N5C7 = CC = CC = C74)O)C (= O)OC | 7.40 | NET1 |
| CCCCCCCCCC[C@H](CC (= O)N[C@@H](CC (= O)N)C (= O)N <br> [C@@H](C(CC (= O)N)O)C (= O)N[C@@H](CC(C)C)C = O)O | 7.38 | NET1 |
| C[C@H]1C (= O)N2[C@@H](C[C@]3([C@@H]2NC4 = CC = CC = C43) <br> C5 = CC6 = C(C = C5)C (= CN6)C[C@H]7C (= O)N8CCC[C@H]8C (= O)N7)C (= O)N1 | 7.37 | NET1 |
| CC(C)C[C@H](CC (= O)NO)C (= O)N1CCC[C@H]1C (= O)N[C@@H](C(C)C)C (= O)O | 7.36 | NET1 |
| CC (= CCN1C2 = C(C = CC (= C2)OC)C3 = C1[C@@H] (N4C (= O) <br> [C@@H]5CCCN5C (= O)[C@]4([C@H]3OC)O)C = C(C)C)C | 7.36 | NET1 |
| CC1(C = CC2 = C(O1)C = CC3 = C2N(C4 = C3[C@H]5[C@]67C(C4(C)C)C[C@@]8 <br> (CCCN8C6 = O) <br> C (= O)N7C9[C@]51C2 = C(C3 = C(C = C2)OC(C = C3)(C)C)[N+] (= C1C([C@H]1C92C (= O) <br> N3CCCC3(C1)C (= O)N2)(C)C)[O-])O)C | 7.34 | NET1 |
| CC(/C = C(\\C)/C = C/C (= O)NO)C (= O)C1 = CC = C(C = C1)N(C)C | 7.30 | NET5 |
| CCCCCCCCC[C@@H]1C[C@@H]([C@H]2CN1O[C@@H]2C3 = CC = CC = C3)O | 7.07 | NET5 |
| C[C@H](CCC (= C)C(C)C)C1CC (= O)N = C2[C@@]1 <br> (CCC3 = C2CC[C@@H]4[C@@]3(CC[C@@H](C4)O)C)C | 6.93 | NET5 |
| C[C@@]12CCN([C@@H]1N(C3 = C2C = C(C = C3)OC (= O)NC)C)C | 6.93 | NET5 |
| C1C(NC (= N1)NCC(C(C(CCO)O)O)NC (= O)CC(CCCN)N)C(CN)(C = O)O | 6.88 | NET5 |
| CCCC[C@@H](C)C[C@@H](C)C (= O)N(C)[C@@H](CC(C)C)C (= O)N[C@@H] <br> ([C@@H](C)O)C (= O)N(C)[C@@H](C(C)C)C (= O)N1C[C@H](C[C@H]1C (= O)N2 <br> [C@H](C = CC2 = O)C)O | 6.83 | NET5 |
| CC1 = C2C3 = C(C = CC2 = CC = C1)C4 = CC[C@@H]([C@]4(CC3)C)[C@H](C)/C = C/ <br> [C@@H](C)C(C)C | 6.82 | NET5 |
| CN(NC (= O)[C@H](CCCN = C(N)N)N)P (= O)(C(C (= O)OC)O)O | 6.82 | NET5 |
| CCCC[C@@H](C)C[C@@H](C)C (= O)N(C)[C@@H](C[C@H](C)CC)C (= O)N <br> [C@@H]([C@@H](C)OC (= O)C)C (= O)N(C)[C@@H](C(C)C)C (= O)N1C[C@H](C[C@H]1C <br> (= O)N2[C@H](C = CC2 = O)C)O | 6.82 | NET5 |
| CC(C)([C@H]1CC2 = C(O1)C = CC (= C2)O)O | 6.80 | NET5 |

## 3.3 Reference structure

The 3CL Protease also known as the main protease, and it has 306 amino acids in a single chain, while the active form of the protein is in a dimeric state. There are three structural domains: I, II, and III, where domains I and II are involved in forming the active site of the protein. Domain III is responsible for forming the dimer. This study used the 3CL protease protein structure collected from the PDB database (PDB ID: 6LU7). This structure has single chain submitted with an inhibitor N3 (N-[(5-METHYLISOXAZOL-3-YL)CARBONYL]ALA-NYL-L-VALYL-N~1~-((1R,2Z)-4-(BENZYLOXY)-4-OXO-1-{[(3R)-2-OXOPYRROLIDIN-3-YL]METHYL}BUT-2-ENYL)-L-LEUCINAMIDE). 3CL protease has a catalytic dyad His[41] and Cys[145]. This inhibitor made direct hydrogen bond (H-bond) interaction with Cys[145] while hydrophobic contact with His[41]. This confirmed the inhibitory action of the co-crystallized

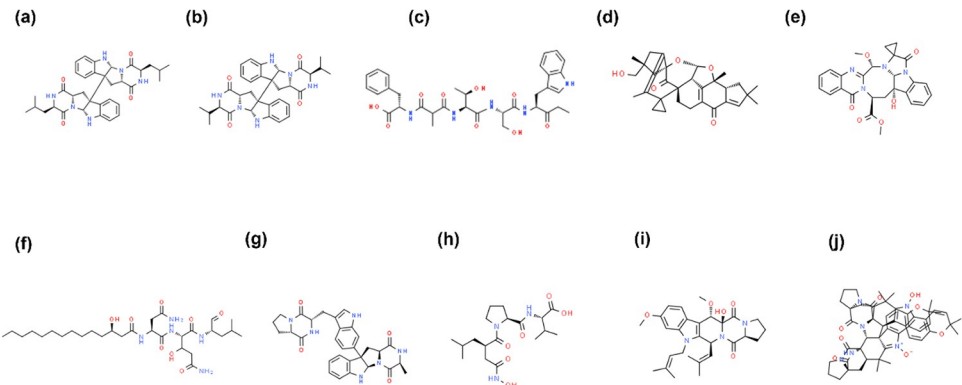

**Fig 2. 2D representation of the top 10 compounds screened using NET1 model based on the predicted pIC$_{50}$.** (a) CMP1$^{(NET1)}$ (b) CMP2$^{(NET1)}$ (c) CMP3$^{(NET1)}$ (d) CMP4$^{(NET1)}$ (e) CMP5$^{(NET1)}$ (f) CMP6$^{(NET1)}$ (g) CMP7$^{(NET1)}$ (h) CMP8$^{(NET1)}$ (i) CMP10$^{(NET1)}$ (j) CMP4$^{(NET1)}$.

molecule N3. Glu$^{166}$ mutation in 3CL protease showed its significant role in the biological activity of the protein [77]. This residue made a hydrogen bond with the N3 inhibitor in its crystal structure as shown in **Fig 4**. Gly$^{143}$ is considered as the most preferred residue for forming H-bond with the ligand molecule along with Cys$^{145}$, and His$^{163}$, and Glu$^{166}$. Inhibitor N3 has H-bonds formed with Gly$^{143}$ and Glu$^{166}$ of the protein, which further indicated its strong binding. In addition, Thr$^{190}$ showed H-bond formation with the co-crystallized inhibitor N3.

In conjunction with N3, which serves as a covalent inhibitor, a reversible non-covalent inhibitor called OEN was employed as an additional reference ligand, sourced from the PDB structure with PDB ID: 7L0D. **S1 Fig** illustrates the interaction plot between OEN and the 3CL protease. The interactions demonstrated by OEN were notably similar to those of N3, with Asn142 and Gly143 representing the two key residues that formed hydrogen bonds with OEN, similar to their direct interaction with N3. N3 had a broader range of interactions due to its extended structure. As a result, this study utilized the N3 interacting residues in the binding site design to allow for a more extensive conformational search.

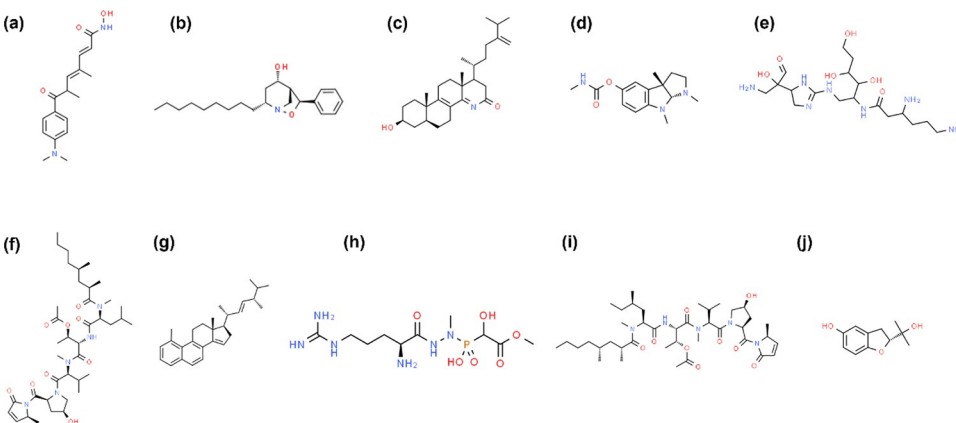

**Fig 3. 2D representation of the top 10 compounds screened using NET5 model based on the predicted pIC$_{50}$.** (a) CMP1$^{(NET5)}$ (b) CMP2$^{(NET5)}$ (c) CMP3$^{(NET5)}$ (d) CMP4$^{(NET5)}$ (e) CMP5$^{(NET5)}$ (f) CMP6$^{(NET5)}$ (g) CMP7$^{(NET5)}$ (h) CMP8$^{(NET5)}$ (i) CMP10$^{(NET5)}$ (j) CMP4$^{(NET5)}$.

**Table 4. Top 20 compounds screened using NET1 and NET5 model, further ranked on QED score.** Top 5 compounds (highlighted grey) were selected for next phase of docking and simulation.

| S No. | Screened Hits | QED Value |
|---|---|---|
| 1 | CMP4(NET5) | 0.84 |
| 2 | CMP10(NET5) | 0.71 |
| 3 | CMP2(NET5) | 0.65 |
| 4 | CMP9(NET1) | 0.64 |
| 5 | CMP4(NET1) | 0.56 |
| 6 | CMP5(NET1) | 0.54 |
| 7 | CMP3(NET5) | 0.5 |
| 8 | CMP7(NET5) | 0.47 |
| 9 | CMP2(NET1) | 0.44 |
| 10 | CMP1(NET1) | 0.42 |
| 11 | CMP7(NET1) | 0.4 |
| 12 | CMP8(NET1) | 0.34 |
| 13 | CMP1(NET5) | 0.27 |
| 14 | CMP10(NET1) | 0.16 |
| 15 | CMP6(NET5) | 0.15 |
| 16 | CMP9(NET5) | 0.14 |
| 17 | CMP5(NET5) | 0.1 |
| 18 | CMP3(NET1) | 0.09 |
| 19 | CMP8(NET5) | 0.06 |
| 20 | CMP6(NET1) | 0.05 |

## 3.4 Hit compounds docking

The 3CL protease structure from 6LU7 was prepared using the AutoDock Tool (ADT) kit that adds hydrogen to the 3D coordinates and Gasteiger charges on each atom of the protein, which is based on the partial equalisation of orbital electronegativity. Further, the top 5 hits shown in Table 4 were also prepared using ADT. The grid box for docking was designed based on the inhibitor N3 position in the 6LU7 structure. Table 5 shows the binding energies calculated by AutoDock Vina for the 20 poses generated for each candidate.

Here, the best docking energy was shown by CMP4(NET1) of -7.1 kcal/mole for the first pose. Followed by this compound, CMP4(NET5) and CMP9(NET1) showed strong binding energy of -6.9 kcal/mole. Other two compounds, CMP10(NET5) and CMP2(NET5) showed relatively poor binding energy in their docked poses, the best docked pose for these compounds showed -5.8 kcal/mole and -5.6 kcal/mole. CMP9(NET1) had the best average binding energies of -6.1 kcal/mole while the second-best average energy was shown by CMP4(NET1) with -6.07 kcal/mole. The best pose was considered for further analysis as it showed the best binding energy. **S2 Fig** shows the 3D and 2D interaction poses of all hits with the protein. Each candidate showed a hydrogen bond except the CMP4(NET5) molecule. CMP10(NET5) formed two H-bond, with Glu[166] and Tyr[54]. Glu[166] that considered as critical active site residue was also involved in the forming H-bond with CMP9(NET1). Gln[189] forms an H-bond with the CMP2(NET5). Finally, CMP4(NET1) formed an H-bond with Asn[142]. Here, it was observed that His[41] from the catalytic dyad was involved in hydrophobic contact with CMP2(NET5), CMP4(NET5) and CMP10(NET5) in its complex.

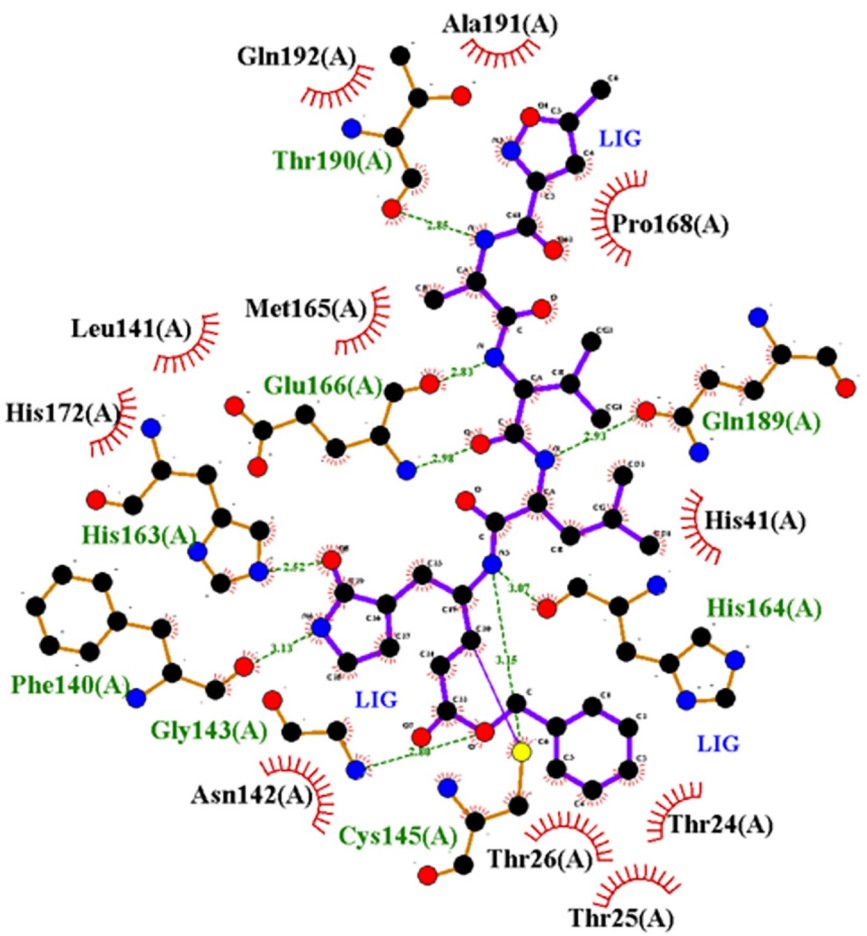

**Fig 4. Interaction plot of native inhibitor N3 with 3CL protease in the protein crystal structure 6LU7.** Hydrogen bonds are shown in the green dashed line. Other residues formed hydrophobic contacts.

### 3.5 Molecular dynamics simulation analysis

In this study, molecular dynamics (MD) simulation was used to estimate the contact intensity of the protein-ligand binding of the selected top five hits. The post-dynamics simulation analysis for protein-ligand complexes gives important information on the system's flexibility. The best docked pose of the selected top hits was used in the MD simulation. The root mean square deviation (RMSD) was calculated over the 100 ns simulation for the five hits to filter out only the stable complexes. Later, only the top two were selected for evaluating additional properties,

**Table 5. Binding energies (kcal/mole) calculated by AutoDock for the top 20 poses generated in docking of top 5 hits with 3CL protease.**

| Compounds | P 1 | P 2 | P 3 | P 4 | P 5 | P 6 | P 7 | P 8 | P 9 | P 10 | P 11 | P 12 | P 13 | P 14 | P 15 | P 16 | P 17 | P 18 | P 19 | P 20 |
|---|---|---|---|---|---|---|---|---|---|---|---|---|---|---|---|---|---|---|---|---|
| CMP4[(NET5)] | -6.9 | -6.5 | -6.4 | -6.2 | -6.2 | -6.1 | -6.1 | -6.1 | -6 | -6 | -5.9 | -5.8 | -5.8 | -5.7 | -5.7 | -5.7 | -5.7 | -5.6 | -5.6 | -5.6 |
| CMP10[(NET5)] | -5.8 | -5.6 | -5.5 | -5.5 | -5.5 | -5.4 | -5.4 | -5.4 | -5.3 | -5.3 | -5.3 | -5.2 | -5.2 | -5.2 | -5.1 | -5.1 | -5 | -5 | -5 | -5 |
| CMP2[(NET5)] | -5.6 | -5.5 | -5.5 | -5.5 | -5.5 | -5.5 | -5.4 | -5.4 | -5.3 | -5.3 | -5.3 | -5.3 | -5.2 | -5.2 | -5.2 | -5.2 | -5.1 | -5.1 | -5.1 | -5.1 |
| CMP9[(NET1)] | -6.9 | -6.8 | -6.7 | -6.7 | -6.7 | -6.5 | -6.3 | -6.2 | -6.1 | -6.1 | -6 | -6 | -6 | -5.8 | -5.8 | -5.8 | -5.8 | -5.7 | -5.7 | -5.7 |
| CMP4[(NET1)] | -7.1 | -6.9 | -6.8 | -6.5 | -6.2 | -6.2 | -6.1 | -6.1 | -6 | -6 | -6 | -5.8 | -5.8 | -5.8 | -5.8 | -5.8 | -5.7 | -5.7 | -5.6 | -5.6 |

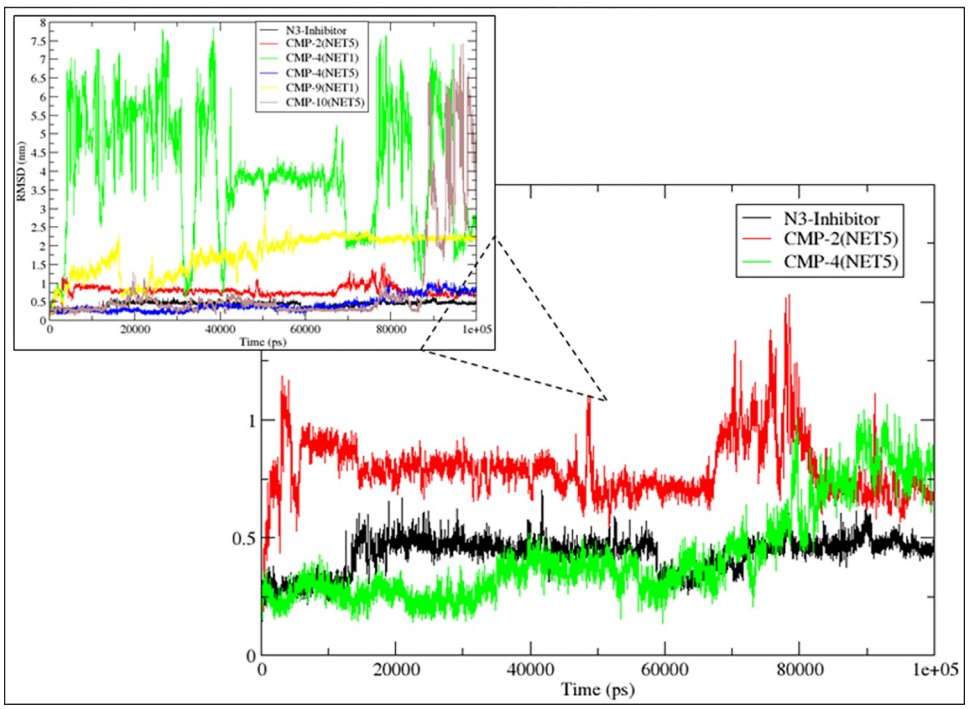

**Fig 5. The RMSD of the ligands calculated over 100 ns MD simulation trajectories for the top five hits (CMP4(NET1), CMP9(NET1), CMP10(NET5), CMP2(NET5), CMP4(NET5)) and the reference ligand N3.** The bigger plot zooms the RMSD of native inhibitor N3, CMP2(NET5) and CMP4(NET5).

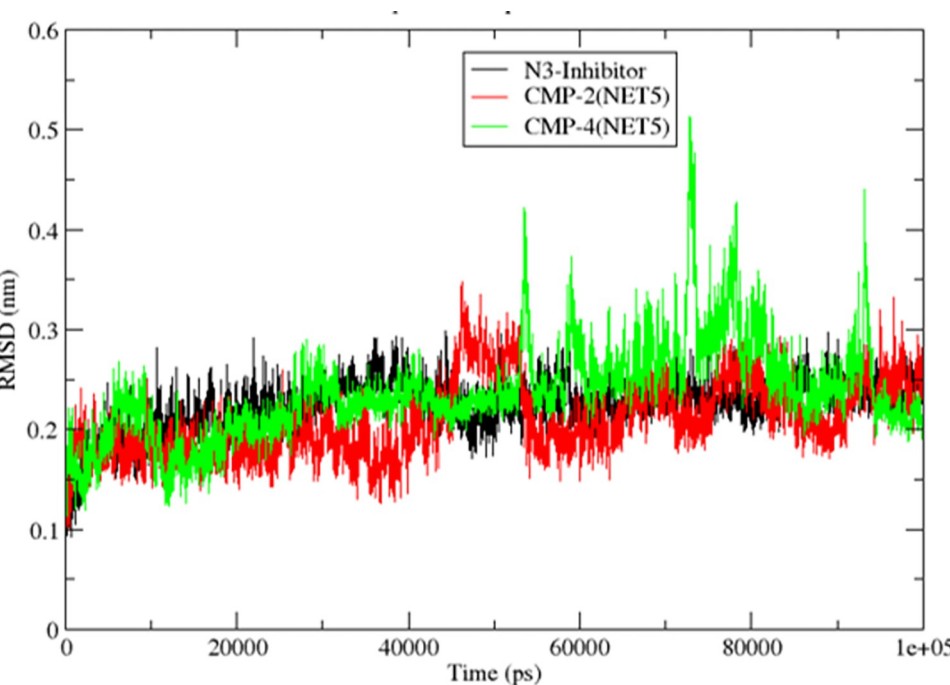

**Fig 6. The RMSD of the protein molecule calculated over 100 ns MD simulation trajectories for the selected hits CMP2(NET5) and CMP4(NET5) and the reference ligand N3.**

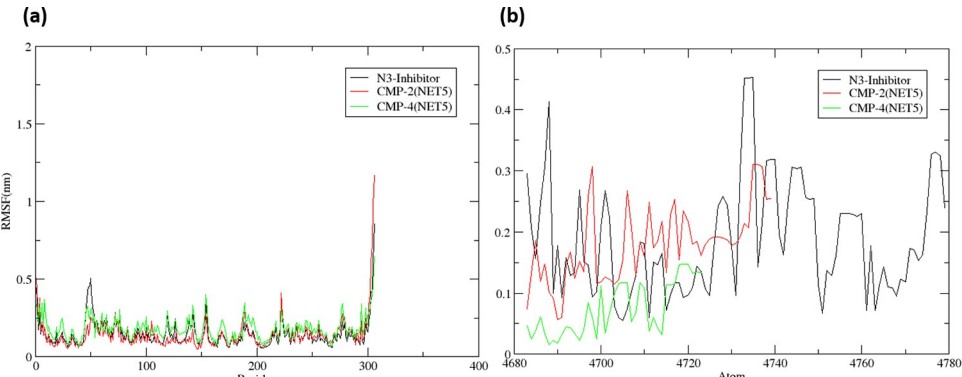

**Fig 7. The RMSF for CMP2$^{(NET5)}$, CMP4$^{(NET5)}$ and reference ligand N3 complexes, calculated over 100 ns MD simulation trajectories for (a) protein and (b) ligands.**

including the root mean square deviation (RMSD), the root mean square fluctuation (RMSF), binding free energy on the MMGBSA protocol, and pull force in steering dynamics.

### 3.5.1 Root Mean Square Deviation (RMSD)

The stability of the compound with respect to binding to the protein was investigated through root mean square deviation (RMSD). The RMSD determined the measure of conformational variation that proteins and ligands undergo upon binding. **Fig 5** shows the RMSD of the ligands when they are in bound state with the protein. Here, it was observed that the compounds CMP2$^{(NET5)}$ and CMP4$^{(NET5)}$ showed stable and consistent conformation with RMSD of ranged from 0.4 nm to 1 nm. Both the compounds had RMSD of 0.5 nm for the last 20 ns simulation. However, the compounds CMP4$^{(NET1)}$, CMP9$^{(NET1)}$, CMP10$^{(NET5)}$ showed high RMSD compared to the other two hits. The RMSD of CMP4$^{(NET1)}$ peaked relatively the highest RMSD value of 8 nm during the 100 ns simulation, while CMP10$^{(NET5)}$ stayed stable with RMSD of 0.4 nm for 85 ns simulation but peaked to 7.5 nm for the rest of the simulation. Here, the compound CMP9$^{(NET1)}$ peaked to 2.5 nm during the 100 ns simulation. **S3 Fig** shows the dissociation of ligand molecules during simulation. The smaller plot in **Fig 5** showed that these compounds (CMP4$^{(NET1)}$, CMP9$^{(NET1)}$, CMP10$^{(NET5)}$) did not exhibit the bound state conformation with the proteins. The reference ligand N3 showed highly stable and consistent RMSD of 0.4 nm to 0.5 nm during the 100 ns simulation. As shown in the bigger plot of **Fig 5**, the compounds (CMP2$^{(NET5)}$ and CMP4$^{(NET5)}$) showed a similar trend of RMSD with the reference ligand N3, therefore they were selected for further analysis. This plot also shows that native inhibitor N3 had a jump in the conformational space for the first 10–12 ns of the simulation but then it stabilized. Visual inspection of this compound verified its large molecular structure, which has a certain scope for rotation. However, no significant translational motion was observed. Similar behaviour was shown by CMP2$^{(NET5)}$ and CMP4$^{(NET5)}$ where the compounds showed high rotational motion that caused RMSD to reach 0.75 nm.

In contrast, the other three ligands CMP4$^{(NET1)}$, CMP9$^{(NET1)}$, and CMP10$^{(NET5)}$ showed very high translational motion and moved out of the binding site. Protein Cα RMSD was also calculated for these two selected compounds and the native inhibitor, shown in Fig 6. RMSD of the protein showed a high consistent behaviour where it ranged under 0.3 nm for most simulation frame. In CMP4$^{(NET5)}$, proteins showed some fluctuation between 70–80 ns time frame. However, it quickly gets stabilized under 0.3 nm as shown in **Fig 6.**

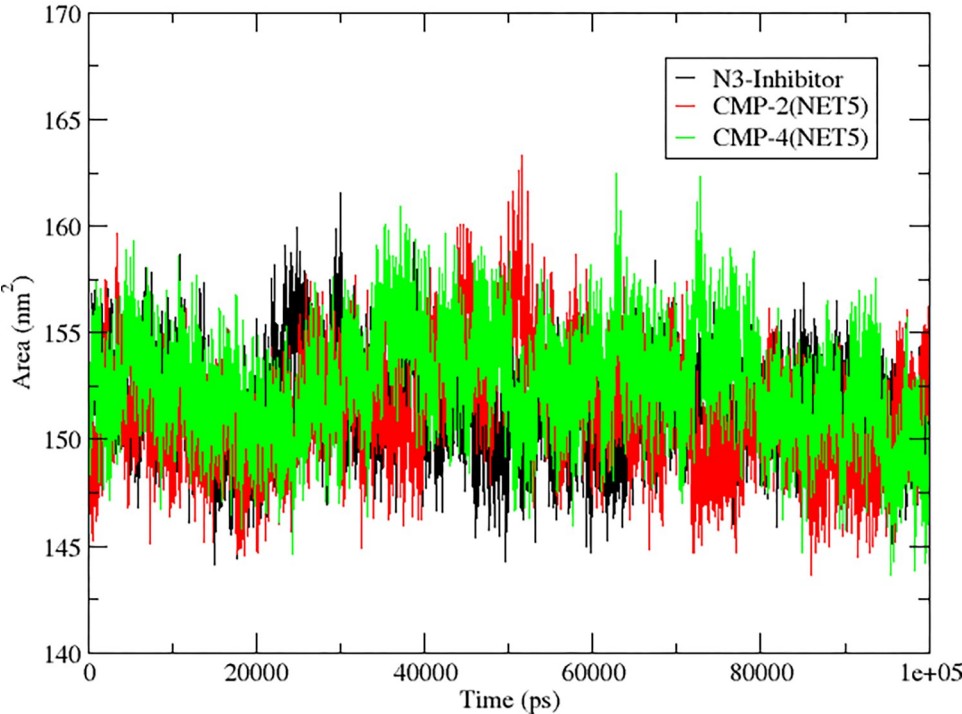

**Fig 8. Solvent accessible surface area (SASA) of protein in bound state with CMP2(NET5), CMP4(NET5) and reference ligand N3 over the period of 100 ns MD simulation.**

NVT and NPT ensemble equilibrium graphs are shown in **S4 Fig**. Temperature was fixed to 310 K and system achieved this during NVT equilibrium. Pressure was 1 bar for the system, and as it can be seen in **S4 Fig**, pressure has fluctuated under the acceptable range. The structural influence of ligand binding has also been estimated by calculating the radius of gyration (Rg) which measures the globularity of the system. Supplementary **S5 Fig** shows the Rg of the protein under three states when it bound with hits CMP2(NET5) and CMP4(NET5) and the reference ligand N3, respectively. Here, the protein structure showed similar globularity trend for all three complexes. However, few peaks were observed for the protein bound with CMP2(NET5) and CMP4(NET5) compounds, but they quickly settled to the initial state. Average Rg of the protein shown in all three complexes was 2.25 nm. **Fig 6**, **S4** and **S5** **Figs** collectively showed that the complexes for these three compounds (1) CMP2(NET5) (2) CMP4(NET5) and (3) reference ligand N3 were stable in 3D space, and both the hits behaved similarly with the reference ligand.

**3.5.2 RMSF analysis.** RMSF values were calculated for the protein and three molecules (CMP2(NET5), CMP4(NET5) and reference ligand N3) after binding to estimate the individual fluctuations of each residue/atom. **Fig 7(A)** shows the RMSF of the protein structure upon binding of the ligands. The RMSF of the protein for the CMP4(NET5) showed maximum peaks during the for 23 residues with RMSF > 0.3 nm. The protein structure bound to the CMP2(NET5) and reference ligand N3 showed similar trend with 10 and 14 residues with RMSF > 0.3 nm. Overall, the RMSF of the proteins showed a similar trend of fluctuations with marginal abruption with peak.

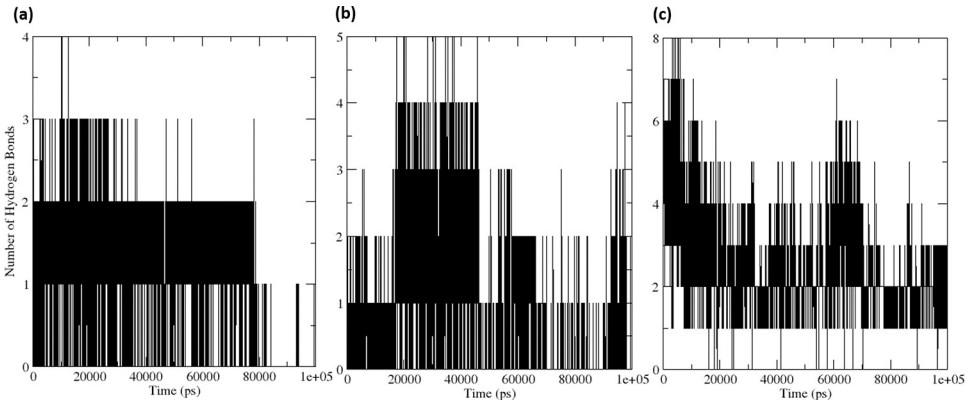

**Fig 9.** Hydrogen bond counts over the 100 ns MD simulation trajectories for the protein-ligand complexes (a) CMP4$^{(NET5)}$ (b) CMP2$^{(NET5)}$ and (c) native inhibitor N3.

The RMSF calculated over each atom for the ligands is shown in **Fig 7(B)**. Here, the RMSF for the reference ligand N3 was found to have a higher number of peaks,14 atoms with RMSF > 0.3 nm compared to others. The CMP2(NET5) showed similar trend, 4 atoms with RMSF > 0.3 nm. However, CMP4(NET5), showed the lowest fluctuation with no atoms with RMSF > 0.3 nm.

**3.5.3 SASA (solvent accessible surface area).** The SASA (solvent accessible surface area) of a protein is the area on its surface that is most proximal to the surrounding solvent and thus exhibits the greatest degree of direct interaction with it. Throughout the 100 ns MD simulation, the SASA values of the protein bound with the CMP2$^{(NET5)}$, CMP4$^{(NET5)}$ and reference ligand N3 were calculated, and the graphs were plotted, as shown in the **Fig 8**. SASA measurements showed that protein in all three complexes had SASA of 148–158 nm$^2$. However, a minor rise to 160 nm$^2$ in SASA was detected in the CMP2$^{(NET5)}$, CMP4$^{(NET5)}$ and reference ligand N3 at the 50 ns, 60 ns and 30 ns, respectively, due to the exposure of internal residues caused by a change in the protein's conformation.

### 3.5.4 Hydrogen bonds

Intermolecular hydrogen bonding can be utilized as a metric to evaluate the degree of protein-ligand binding as well as the stability of the complex. During a 100 ns simulation, the total number of hydrogen bonds formed by three compounds ranged from 1 to 8, as depicted in **Fig 9**. The native inhibitor N3 exhibited 2–3 hydrogen bonds with high fluctuations and 3–6 hydrogen bonds in a stable configuration. **Fig 9(C)** illustrates two frames, one from 0 ns to 10 ns and the other from 10 ns to 20 ns, where 3–6 hydrogen bonds and 2–4 hydrogen bonds were detected in the protein-ligand complex of the native inhibitor N3. CMP4$^{(NET5)}$ formed 1–2 hydrogen bonds with the binding pocket residues of the protein with minimal fluctuation, and 0–1 hydrogen bonds with high fluctuation, as shown in **Fig 9(A)**. Furthermore, CMP2$^{(NET5)}$ displayed 0–1 hydrogen bonds with high fluctuations and 1–3 hydrogen bonds with minimal fluctuation during the 100 ns MD simulation, as depicted in **Fig 9(B)**. The native inhibitor N3-protein complex formed the highest number of hydrogen bonds during the simulation in comparison to the other two top hits. Additionally, it was observed that CMP2 (NET5) demonstrated the consecutive highest number of hydrogen bonds.

**3.5.5 MD simulation protein-ligand interaction.** Later, the complexes formed in the simulation were collected at different timeframe to read the positional and interaction variability. As it was observed in the RMSD plot (**Fig 6**) the native inhibitor is stabilized throughout

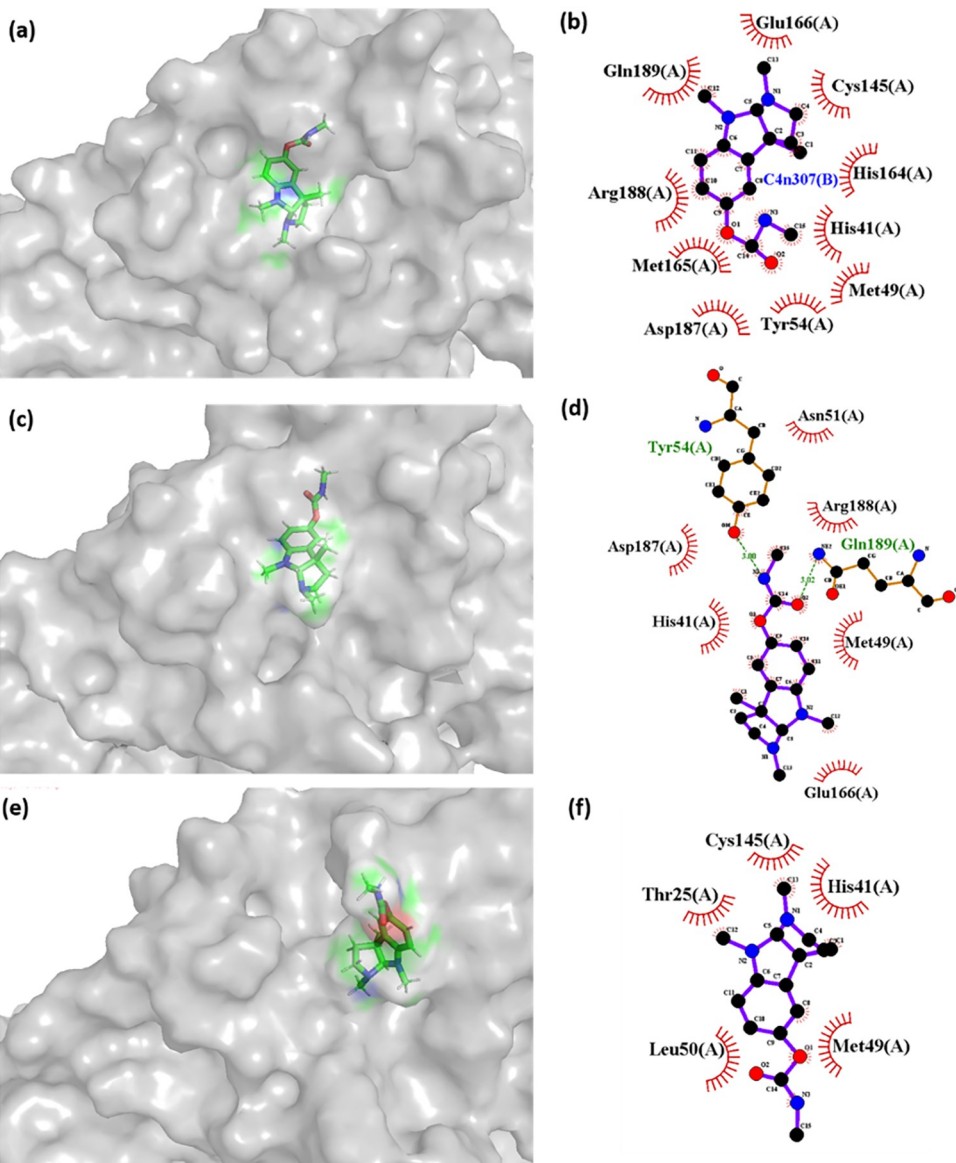

**Fig 10.** 3D and 2D interaction plot of CMP4^(NET5) with the 3CL protease protein at (a-b) 0 ns (c-d) 50 ns, and (e-f) 90 ns of the simulation trajectory.

the simulation. However, CMP2^(NET5) showed conformational stability from 10–70 ns simulation time and later from 80–100 ns. Thus, three structures were extracted from the simulation trajectory at 0 ns (initial state), 50 ns (first stable zone), and 90 ns (second stable zone). Similarly, CMP4^(NET5) also had similar stable time zones, and their three structures were also extracted from the trajectory at 0 ns, 50 ns, and 90 ns. Native inhibitor N3 simulation trajectory was also treated similarly to match with the hit compounds. However, in the native inhibitor, there is only one single stable zone (10–100 ns). **Figs 10, 11 and S6 Fig** shows the 3D and 2D interaction plot of protein-ligand at 0, 50, and 90 ns respectively.

Native inhibitor N3 interaction plot for the poses generated at 0, 50 and 90 ns are shown in **S6 Fig**. Here, the highest number of H-bonds is shown in the first pose (0 ns). In this pose, six residues were involved in H-bonding, they are: His[163], His[164], Gly[143], Gln[189], Glu[166], and

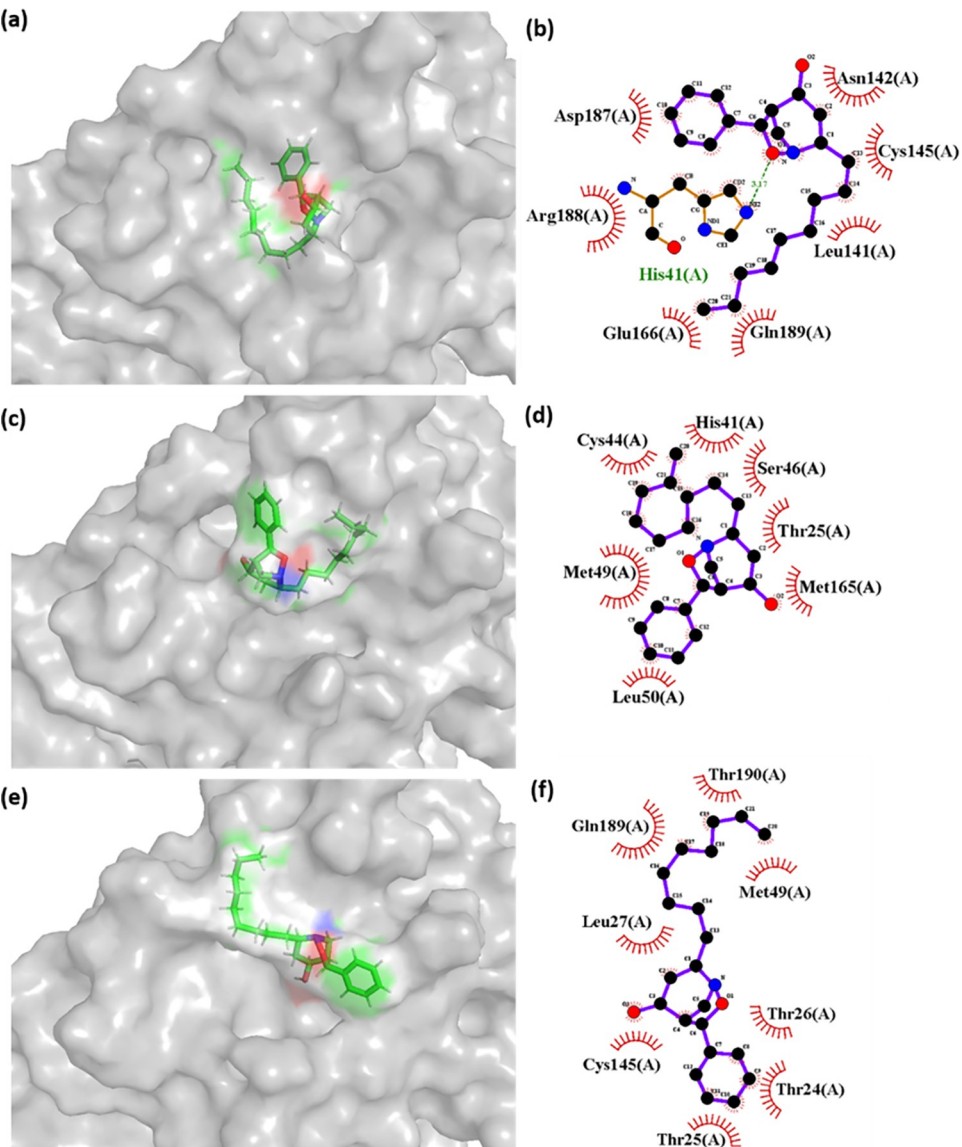

**Fig 11.** 3D and 2D interaction plot of CMP2[(NET5)] with the 3CL protease protein at (a-b) 0 ns (c-d) 50 ns, and (e-f) 90 ns of the simulation trajectory. 2D interaction map was formed using LigPlus.

Thr[190]. Both catalytic residues were found in the interacting range. However, when structure moved to 50 ns, H-bonds were reduced to two where Gln[189] was from the earlier list while Thr26 was added as new H-bond forming residue. Pose collected at 90 ns showed highly similar interaction behaviour as 50 ns pose.

**Fig 10** shows the interactions detected in the complex of CMP4[(NET5)] with 3CL protease protein in 3D and 2D formats. In CMP4[(NET5)], the first pose at 0 ns did not show any polar contact. However, His[41] and Cys[145] were observed in the interaction plot under interacting range. Another critical residue of 3CL protease, Glu[166] was also found in the interacting vicinity. Later, at 50 ns, Cys[145] got disappeared, but additional H-bond formed with Gln[189] and Tyr[54]. Moreover, His[41] was still there in the interacting zone. This shows the high possibility of CMP4[(NET5)] interacting with His[41] either in hydrophobic contact or in H-bond. Eventually, at 90 ns the H-bonds lost but Cys[145] appeared in the neighbourhood of the compound.

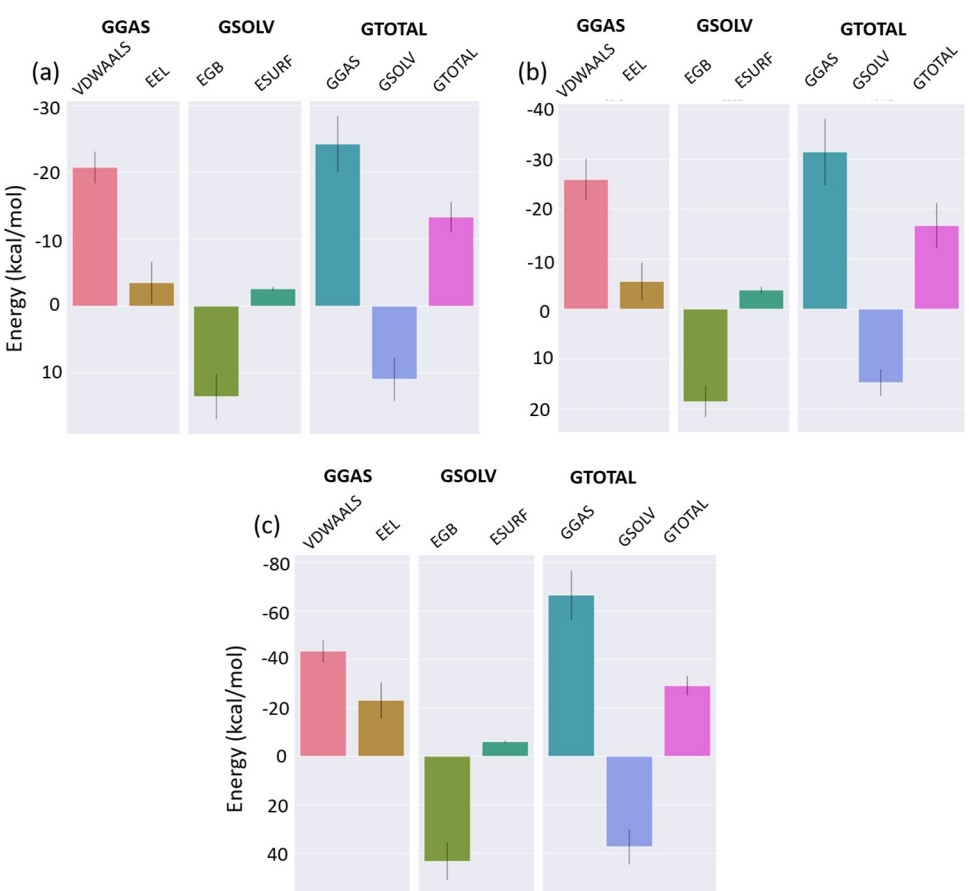

**Fig 12.** MM/GBSA binding free energies for the (a) CMP4[(NET5)] (b) CMP2[(NET5)] and (c) native inhibitor N3, various energetic components are shown in different colour. Gmx MMGBSA tool was used for plotting.

Consistent proximity of CMP4[(NET5)] shows its high interaction probability with the catalytic dyad, which can lead to the protein's activity inhibition. **Fig 10** showed that compound conformation at 90 ns was significantly different compared to 0 and 50 ns poses. Compound orientation was majorly shifted in this region, and the same was shown in the RMSD plot of this compound.

**Fig 11** shows the interaction details of CMP2[(NET5)], His[41] was involved in forming the H-bond in the pose formed at beginning of the simulation (0 ns). Cys[145] was also observed in the interaction range. This confirmed the presence of a catalytic dyad in the interaction range of CMP2[(NET5)]. Moreover, Glu[166] which considered as critical residue for 3CL protease was also marked in the interaction map at 0 ns of simulation. Other two poses at 50 ns and 90 ns were devoid of H-bond. However, His[41] and Cys[145] were found at 50 ns and 90 ns respectively.

**Table 6. Clustering result for last 20 ns time frame of MD Simulation for the top two hits and the native inhibitor of 3CL protease.**

| Compound | Number of Clusters | Population (number of structure) |
|:---:|:---:|:---:|
| 3CL protease | 1 | 2001 |
| CMP2[(NET5)] | 1 | 2001 |
| CMP4[(NET5)] | 1 | 2001 |

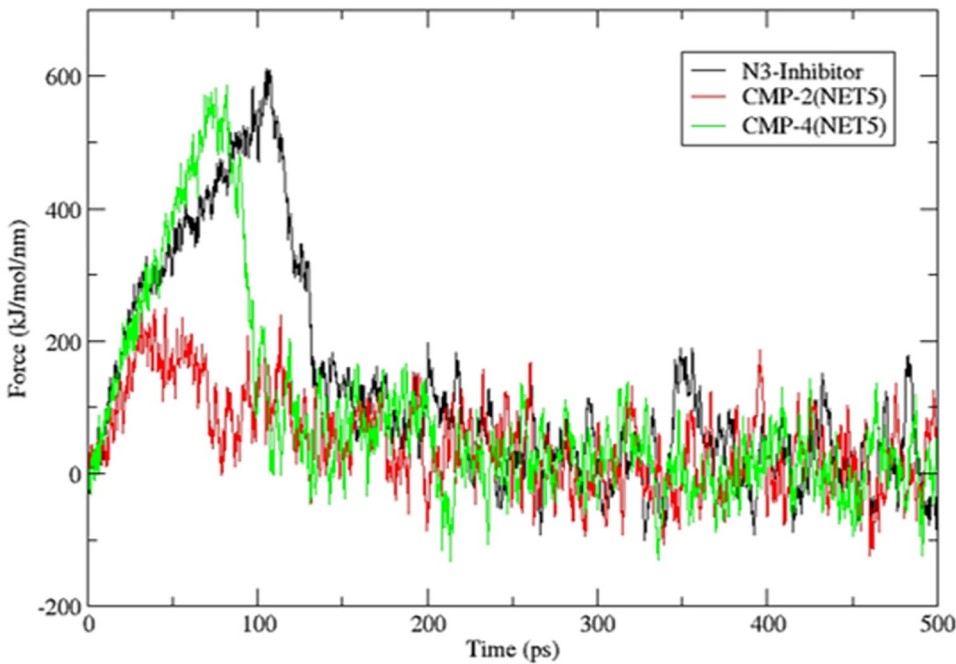

**Fig 13. Force on harmonic spring showed by three complexes (one native inhibitor and two hit compound) during the 500 ps timescale od steered MD simulation.**

Structure, formed at 50 ns was extracted from the most stable zone of the simulation for both hit compounds. In this poses (50 ns), CMP4(NET5) showed stronger interaction and involvement of catalytic residues compared to CMP2(NET5). 3D depiction of both hits shows their binding at a similar binding site, but rotational motion within the molecule (resulting from the degree of freedom) allowed them to change their conformations.

**3.5.6 MD simulation protein-ligand interaction.** Binding free energies (ΔG) for all three complexes CMP4(NET5), CMP2(NET5) and the reference ligand N3 were calculated for the last 20 ns of the simulation and averaged to estimate the overall binding strength. **Fig 12** showed the binding free energies (ΔG) of CMP4(NET5), CMP2(NET5) and the reference ligand N3. Native inhibitor showed the most minimum $\Delta G_{Total}$ that composed of different components mentioned in method section. Electrostatic and van der Waal showed the best performance in stabilizing the complex. Average electrostatic energy in native inhibitor complex was –23.11 kcal/mole while the van der Waal energy was -43.43 kcal/mole as shown in **Fig 12(C)**. This made the overall binding energy -29.18 kcal/mole after adding the solvation term of 37.36 kcal/mole. Moreover, CMP4(NET5), CMP2(NET5) compounds showed similar ΔG binding free energies. Their total ΔG binding energies were -13.32 kcal/mole (**Fig 12(A)**) and -16.71 kcal/mole (**Fig 12(B)**) for, CMP4(NET5) and CMP2(NET5), respectively. This showed that hit compounds had acceptable range of ΔG and formed a stable complex with the 3CL protease protein. However, none of the compounds showed better binding strength compared to the native inhibitor N3.

## 3.6 Steered MD simulation

Later, steered MD simulation was carried out to estimate the dissociation magnitude for all the three complexes. The starting structure for the steered MD simulation was obtained from the clustering over the trajectory resulting from the classical MD simulation that was performed

earlier. Table 6 shows the clustering results for the last 20 ns, all three compounds, including the native inhibitor, showed only one cluster formed at the RMS cut-off of 0.3 nm. The central structure of these clusters was used as starting co-ordinate in steered dynamics. In the steered dynamic, compounds dissociated from the protein over the period of 500 ps where the poses were saved after every 1 ps. Initially, there was minor displacement of the ligand molecule from the binding site of the protein and then force of the spring reached to the restoring force within the protein-ligand complex. This is represented as the peak in **Fig 13** that shows the force on the spring in the steered dynamics. Native inhibitor had the maximum resistance (610.57 kJ/mol/nm) to the dissociation as shown by the highest peak in **Fig 13**. CMP4$^{(NET5)}$ also showed comparable resistance (586.04 kJ/mol/nm) for the dissociation and showed the similar peak as native inhibitor. However, it reached the dissociation state earlier than the native inhibitor. In contrast, CMP2$^{(NET5)}$ did not show high restoring force in the complex state, which showed relatively easier dissociation from the protein molecule. **Fig 13** shows the high binding of the native inhibitor and CMP4$^{(NET5)}$ compared to CMP2$^{(NET5)}$. Moreover, the close behaviour of CMP4$^{(NET5)}$ with native inhibitors makes it a promising hit candidate.

## 4. Conclusions

Currently, the world is experiencing periodic peaks in COVID-19 instances. That demands for a therapeutic molecule with the minimum toxic effect that can inhibit essential protein of SARS-CoV-2. 3CL protease of SARS-CoV-2 has been established as a potential drug target and the structure of the protein has also been solved, which catalyze the structure-based drug design. In this perspective, this study demonstrated an application of machine learning combined with a physics-based simulation technique to identify efficient inhibitor compounds against 3CL protease. A natural compound library was screened, and the top-ranked candidates were validated using sophisticated computational techniques. These compounds (CMP2 and CMP4) have shown promising results in the *in-silico* study and can be explored via *in-vitro* and *in-vivo* experiments.

## Supporting information

**S1 Fig. Interaction plot of reversible non-covalent 3CLpro inhibitor, 0EN with 3CL protease in the protein crystal structure 7L0D.** Hydrogen bonds are shown in the green dashed line. Other residues formed hydrophobic contacts.
(TIF)

**S2 Fig. 3D and 2D interaction plots for the top five hits ranked from the list of ten compounds selected from NET1 and NET5.** Plots are shown for the best pose generated after the molecular docking.(a, b) CMP4$^{(NET5)}$ (c, d) CMP10$^{(NET5)}$ (e, f) CMP2$^{(NET5)}$ (g, h) CMP9$^{(NET1)}$ (i, j) CMP4$^{(NET1)}$.
(TIF)

**S3 Fig. Pictorial representation of CMP4$^{(NET1)}$, CMP9$^{(NET1)}$, CMP10$^{(NET5)}$ unbound state with the protein.**
(TIF)

**S4 Fig. NVT and NPT equilibrium of protein ligand complex for CMP2$^{(NET5)}$ and CMP4$^{(NET5)}$ and the reference ligand N3 for temperature and pressure.**
(TIF)

**S5 Fig. Radius of gyration for the protein in bound states with CMP2$^{(NET5)}$ and CMP4$^{(NET5)}$ and the reference ligand N3.**
(TIF)

**S6 Fig.** 3D and 2D interaction plot of native inhibitor N3 with the 3CL-protease protein at (a, b) 0 ns (c, d) 50 ns, and (e, f) 90 ns of the simulation trajectory. 2D interaction map was formed using LigPlus.
(TIF)

## Acknowledgments

The authors extend their appreciation to the Deanship of Scientific Research at King Khalid University, Abha, KSA and Growdea Technologies Pvt Ltd, Gurugram, India for assisting in the bioinformatics work.

## Author Contributions

**Conceptualization:** Md. Zeyaullah, Nida Khan, Khursheed Muzammil, Mohammad Suhail Khan, Wajihul Hasan Khan.

**Data curation:** Md. Zeyaullah, Nida Khan, Khursheed Muzammil, Abdullah M. AlShahrani, Razi Ahmad, Wajihul Hasan Khan.

**Formal analysis:** Md. Zeyaullah, Nida Khan, Khursheed Muzammil, Abdullah M. AlShahrani, Wajihul Hasan Khan.

**Funding acquisition:** Abdullah M. AlShahrani.

**Investigation:** Wajihul Hasan Khan.

**Methodology:** Md. Zeyaullah, Nida Khan, Wajihul Hasan Khan.

**Project administration:** Md. Shane Alam, Wajihul Hasan Khan.

**Resources:** Nida Khan, Md. Shane Alam.

**Software:** Md. Zeyaullah, Nida Khan, Khursheed Muzammil, Md. Shane Alam, Razi Ahmad, Wajihul Hasan Khan.

**Supervision:** Nida Khan, Razi Ahmad, Wajihul Hasan Khan.

**Validation:** Nida Khan, Mohammad Suhail Khan, Razi Ahmad.

**Visualization:** Mohammad Suhail Khan, Razi Ahmad.

**Writing – original draft:** Md. Zeyaullah, Nida Khan, Razi Ahmad, Wajihul Hasan Khan.

**Writing – review & editing:** Khursheed Muzammil, Abdullah M. AlShahrani, Mohammad Suhail Khan, Razi Ahmad, Wajihul Hasan Khan.

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
