## [Decision Letter · Decision Letter 0]

20 Feb 2023

PONE-D-23-03061In-silico Approaches for Identification of Compounds Inhibiting SARS-CoV-2 3CL ProteasePLOS ONE

Dear Dr. Khan,

Thank you for submitting your manuscript to PLOS ONE. After careful consideration, we feel that it has merit but does not fully meet PLOS ONE’s publication criteria as it currently stands. Therefore, we invite you to submit a revised version of the manuscript that addresses the points raised during the review process.

We look forward to receiving your revised manuscript.

Kind regards,

Ahmed A. Al-Karmalawy, Ph.D.

Academic Editor

PLOS ONE

Journal Requirements:

"No".

3. Please clarify the Table numbers uploaded in your manuscript.

Reviewers' comments:

Reviewer's Responses to Questions

**Comments to the Author**

1. Is the manuscript technically sound, and do the data support the conclusions?

Reviewer #1: Yes

Reviewer #2: Yes

Reviewer #3: Yes

2. Has the statistical analysis been performed appropriately and rigorously? 

Reviewer #1: Yes

Reviewer #2: Yes

Reviewer #3: Yes

3. Have the authors made all data underlying the findings in their manuscript fully available?

Reviewer #1: Yes

Reviewer #2: Yes

Reviewer #3: Yes

4. Is the manuscript presented in an intelligible fashion and written in standard English?

Reviewer #1: Yes

Reviewer #2: Yes

Reviewer #3: Yes

5. Review Comments to the Author

Reviewer #1: The authors provided a computational approach for identifying promising inhibitors, of natural origin, against the SARS-CoV-2 3CLpro biotarget. They adopted combined deep learning machine models molecular docking-coupled dynamics simulation which identified two promising hits as reversible inhibitors for the virus main protease enzyme. Additionally, the authors performed steered molecular dynamics simulation ensuring stability of one hit depicting dissociation resistance mimicking the co-crystallized ligand. This manuscript is relevant, valuable in the field, and with potentiality for high citation. However, there are some points should be considered prior publication.

• Comments:

1. The authors adopted the PDB.file (6LU7) for performing the computational analysis which is deposited in its monomeric state. Typically, the 3CLpro is a homodimer relating to its biological activity (10.1126 / science.abb3405) and so computational study should have been performed in its homodimeric state since it is the active form of the protein. Additionally, a comparative data analysis of each 3CLpro protomer could be provided for gaining greater insights regarding the effect of ligand binding on dimerization since the dimerization state is proximity with the 3CLpro canonical binding site. It is advised that the authors would discuss this, while elaborating on the adequacy of computational screening on the 3CLpro dimer versus monomer states.

https://doi.org/10.1038/s41598-021-88630-9

https://doi.org/10.1080/07391102.2021.1880481

2. The co-crystalline ligand (N3) at the 6LU7 PDB.file has been reported as covalent inhibitor for the virus protein. This Michael acceptor (peptidyl) inhibitor was used by the authors for developing models that used to screen libraries of reversible acting drugs. It is recommended to validate the obtained models on other PDB.file with reversible non-covalent 3CLpro inhibitor (e.g. PDB ID: 7L0D).

3. Similarly, the N3 is a proteinomimetic drug that is large and extended across the 3CLpro binding site. Therefore, it was expected that large-sized hits rather than small ligands would predict better fit at the obtained models. However, this approach could miss small molecules with potential 3CLpro activity. Thus, authors should elaborate more how their model could avoid such bias.

4. Authors mentioned the analysis of RMSF matrix within the experimental section, however, no data were presented. Monitoring the RMSF of the bound and apo target proteins in relation to their alpha-carbon references provided further stability analysis by dissecting the protein's flexibility/immobility profiles down to their constituting residues. Authors should report RMSF data so as to grasp the residue-wise dynamic behaviors at the protein’s binding pocket/vicinal loops in addition to pinpointing the key amino acids being significant for the ligand’s anchoring.

Reviewer #2: Manuscript title: In-silico Approaches for Identification of Compounds Inhibiting SARSCoV-2 3CL Protease (Manuscript ID: PONE-D-23-03061).

This manuscript focused on finding an in-silico cure for COVID-19 using several tools, such as machine learning techniques with virtual drug screening methods. This manuscript is likely to meet the basic requirements of the journal. However, the manuscript is not mature, and the points should be addressed. **Please check for the attached reviewer's 2 file.**

Reviewer #3: The author reported in this manuscript “In-silico Approaches for Identification of Compounds Inhibiting SARS-CoV-2 3CL Protease” describes the molecular modeling of set of potent compounds able to exhibit the 3CL protease protein of SARS-CoV-2 and estimate the molecular docking scores and simulations using plenty of models. The paper is written very well and explained in an organized way and the applications of machine learning with some physico chemical based model to identify efficient inhibitors against 3CL-protease.please address the following questions in the manuscript. I am recommending for the publication of this article after the following recommended changes in the manuscript:

1. In page 26, figure 2, from CMP5 until CMP10 they have all symbol (C) while all compounds into the figure has symbols from (a) to (j), Please correct it.

2. In page 27, it is the same as figure 2 , all copunds into the figure has symbols from (a) to (j) while into the caption most of them has symbol (C)

3. Clarify what is the violet colors into the table 1

6. PLOS authors have the option to publish the peer review history of their article (what does this mean?). If published, this will include your full peer review and any attached files.

Reviewer #1: No

Reviewer #2: **Yes: **Haytham O. Tawfik

Reviewer #3: **Yes: **Hazem Essam Okda

---

## [Author Response · Author response to Decision Letter 0]

17 Mar 2023

Manuscript title: In-silico Approaches for Identification of Compounds Inhibiting SARS-CoV-2 3CL Protease (Manuscript ID: PONE-D-23-03061).

Reviewer #1: The authors provided a computational approach for identifying promising inhibitors, of natural origin, against the SARS-CoV-2 3CLpro biotarget. They adopted combined deep learning machine models molecular docking-coupled dynamics simulation which identified two promising hits as reversible inhibitors for the virus main protease enzyme. Additionally, the authors performed steered molecular dynamics simulation ensuring stability of one hit depicting dissociation resistance mimicking the co-crystallized ligand. This manuscript is relevant, valuable in the field, and with potentiality for high citation. However, there are some points should be considered prior publication.

•Comments:

1. The authors adopted the PDB.file (6LU7) for performing the computational analysis which is deposited in its monomeric state. Typically, the 3CLpro is a homodimer relating to its biological activity (10.1126 / science.abb3405) and so computational study should have been performed in its homodimeric state since it is the active form of the protein. Additionally, a comparative data analysis of each 3CLpro protomer could be provided for gaining greater insights regarding the effect of ligand binding on dimerization since the dimerization state is proximity with the 3CLpro canonical binding site. It is advised that the authors would discuss this, while elaborating on the adequacy of computational screening on the 3CLpro dimer versus monomer states.

https://doi.org/10.1038/s41598-021-88630-9

https://doi.org/10.1080/07391102.2021.1880481

Response: Thank you for the suggestions the reviewer has made. In 3CLpro, the monomeric form is the precursor to the dimeric form. The enzyme is initially synthesized as a monomer, which then dimerizes to form the active enzyme complex. After synthesis, the monomeric 3CLpro molecules are thought to undergo a conformational change that enables them to dimerize. Therefore, targeting the monomeric form of 3CLpro may stop the formation of the active form of the enzyme. The explanation of the why the monomeric form of 3CLpro has been used in this study was included in the section “Introduction” of the revised manuscript. Kindly review the lines starting with “3CLpro is known to exist in a homodimeric ….” 

2. The co-crystalline ligand (N3) at the 6LU7 PDB.file has been reported as covalent inhibitor for the virus protein. This Michael acceptor (peptidyl) inhibitor was used by the authors for developing models that used to screen libraries of reversible acting drugs. It is recommended to validate the obtained models on other PDB.file with reversible non-covalent 3CLpro inhibitor (e.g. PDB ID: 7L0D).

Response: In accordance with the recommendation made by the reviewer, we have included the 0EN interaction as an additional reference ligand. 0EN and N3, on the other hand, displayed interaction plots that were very similar. Kindly review the highlighted version of section 3.3 in the revised manuscript for the explanation of the reasoning behind choosing the N3.

3. Similarly, the N3 is a proteinomimetic drug that is large and extended across the 3CLpro binding site. Therefore, it was expected that large-sized hits rather than small ligands would predict better fit at the obtained models. However, this approach could miss small molecules with potential 3CLpro activity. Thus, authors should elaborate more how their model could avoid such bias.

Response: In this study, we first used the pre-existing DeePurpose models to estimate the IC50 value of a compound. To identify the best model, we ran several models with different parameters on bioactive anti-virus compounds, as listed in Table 1. The selection process was unbiased and did not consider the N3 compound at this stage, as the models were tested on a wide range of chemical scaffolds. Next, we applied the best model to a library of natural compounds to identify potential hits based on their predicted IC50 values, as shown in Table 3. During this stage, we did not consider the activity of N3 in ranking the compounds, and they were selected solely based on their chemical descriptors and comptatilibity with the protein. We only used N3 to verify the interactions of the top hits, as these interactions are necessary to inhibit the activity of 3CLpro. So, we can say that screening method is not biased towards N3 inhibitor.

4. Authors mentioned the analysis of RMSF matrix within the experimental section, however, no data were presented. Monitoring the RMSF of the bound and apo target proteins in relation to their alpha-carbon references provided further stability analysis by dissecting the protein's flexibility/immobility profiles down to their constituting residues. Authors should report RMSF data so as to grasp the residue-wise dynamic behaviors at the protein’s binding pocket/vicinal loops in addition to pinpointing the key amino acids being significant for the ligand’s anchoring.

Response: Thank you for the suggestions. We have included the SASA,RMSF and Hydrogen bonds plot for the 100-ns molecular dynamics simulation. Kindly review sections – 3.5.2 to 3.5.4 of the revised manuscript. 

Reviewer #2: Manuscript title: In-silico Approaches for Identification of Compounds Inhibiting SARSCoV-2 3CL Protease (Manuscript ID: PONE-D-23-03061).

This manuscript focused on finding an in-silico cure for COVID-19 using several tools, such as machine learning techniques with virtual drug screening methods. This manuscript is likely to meet the basic requirements of the journal. However, the manuscript is not mature, and the points should be addressed. Please check for the attached reviewer's 2 file.

This manuscript focused on finding an in-silico cure for COVID-19 using several tools, such as machine learning techniques with virtual drug screening methods. This manuscript is likely to meet the basic requirements of the journal. However, the manuscript is not mature, and the following points should be addressed.

1. The researchers should number the pages of the manuscript for ease of review.

Response: Thanks for the suggestion. We have address this issue in the revised manuscript.

2. The way you write the word “figure” in the text should be unified (Figure or figure) and bold.

Response: Corrections were incorporated in the manuscript as suggested by the reviewer. 

3. The way you write the “IC50 and pIC50” in the text should be corrected (IC50 and pIC50).

Response: Thanks for the suggestion. We address this issue in the revised manuscript.

4. The way you write the word “SARS-COV2” in the text should be unified (SARS-COV2 or SARS-COV-2).

Response: Thanks for the suggestion. Corrections were incorporated in the manuscript.

5. The way you write the word “CMP2(NET5)” in the text should be unified (CMP2(NET5)or CMP-2(NET5)) and the remaining CMPs.

Response: Thanks for the suggestion. We addressed this issue in the revised manuscript.

6. Why do the authors express the simulation duration in figures in Pico and not in Nano, as mentioned in the text?

Response: Thank you for the question. Typically in GROMACS packages, MD simulations use time steps in the order of picoseconds (ps) to express the results. However, these units are expressed in huge numeric values, therefore picoseconds were converted to nanoseconds (ns) . The timeframe has been expressed in the explanation as nanoseconds for the convenience of understanding the timescale correlation with the analysis components of MD simulation including RMSD, RMSF, SASA, Hydrogen bonds. Most importantly picosecond depiction in plots can allow for closure examination for the reader. 

7. The way you write the word “3CL protease” in the text should be unified (3CL protease or 3CL Protease).

Response: Thanks for the suggestion. Corrections were incorporated in the manuscript.

8. The numbers of tables in the first paragraph below "3.4 Hit Compounds Docking" should be corrected as Table 4 and Table 5, not Table 3 and Table 4.

Response: Thanks for the suggestion. Corrections were incorporated in the manuscript.

9. Before the title "3.5 Molecular Dynamics Simulation Analysis" the corrected amino acid was Gln189, not Glu189.

Response: Thank you for the suggestion. Corrections were made in the manuscript.

10. Before title "3.5 Molecular Dynamics Simulation Analysis", His41 was present in CMP2(NET5), CMP4(NET5) and CMP10(NET5) not only in CMP2(NET5).

Response: Thanks for the suggestion. Corrections were made in the manuscript.

10. The distance before the citation number was sometimes deleted and sometimes not. Uniform.

Response: Thanks for the suggestion. Corrections were incorporated in the manuscript.

11. The way you write the word “antiviral” in the text should be unified (antiviral or anti-viral).

Response: Thanks for the suggestion. Corrections were incorporated in the manuscript.

12. The authors should mention the SASA, RMSF, histograms indicating the fractions of binding between the protein’s amino acids and their compounds and the numbers of hydrogen bonds that were seen in the complex formed by compounds during the 100-ns molecular dynamics simulation.

Response: Thank you for the suggestions. We have included the SASA,RMSF and Hydrogen bonds plot for the 100-ns molecular dynamics simulation. Kindly review sections– 3.5.2 to 3.5.4 of the revised manuscript. 

13. What is the meaning of SNMILES in the title of Table 3.?

Response: Corrections were incorporated in the manuscript as suggested by the reviewer. 

14. The caption of Figures 2 and 3 needs correction inside it about the codes of compounds.

Response: The corrections were made according to the suggestion.

15. The letter "N" at the beginning IUPAC name of the native inhibitor should be in italic.

Response: Corrections were incorporated in the manuscript.

16. Below Figure 7., the authors mentioned only five residues involved in H-bonding. Where is the sixth one? (His163).

Response: The manuscript has been corrected according to the suggestion.

17. Below Figure 7., the corrected amino acid is Gln189, not Glu189.

Response: Thanks for the suggestion. Corrections were made in the manuscript.

18. Before Figure 8., in CMP4(NET5) at 50 ns, Glu166 didn’t disappeared.

Response: Thank you for suggestion, correction was made in the revised manuscript.

19. In Figure 8. (b), the amino acid Cys145 didn’t seem clear.

Response: The Figure 8(b) has been updated to Figure 11 and has been modified according to the suggestion made by the reviewer in the revised manuscript.

20. Figure 9 didn’t mention in the text of the manuscript.

Response: The Figure 9 has been updated to Figure 12 and it has been mentioned in the text of the revised manuscript under 3.5.6 section.

21. The table number at "Steered MD Simulation" is Six, not Five. (Table 5 was duplicated).

Response: The correction has been made according to the suggestions. The duplicate Table 5 has been modified to Table 6 in the revised manuscript.

22. The resolution of Figure 9 is not suitable for publication.

Response: The Figure 9 has been updated to Figure 12 in the revised manuscript and the resolution has been improved.

23. The references style in the "reference section" must be uniform and should mention the recent references. The interval of pages should be mentioned in all references.

Response: Thanks for the suggestion. We have corrected the manyscript file and we have added some recent manuscript of the similar topic. 

Finally, the MS will be suitable for publication after the above corrections/changes.

Reviewer #3: The author reported in this manuscript “In-silico Approaches for Identification of Compounds Inhibiting SARS-CoV-2 3CL Protease” describes the molecular modeling of set of potent compounds able to exhibit the 3CL protease protein of SARS-CoV-2 and estimate the molecular docking scores and simulations using plenty of models. The paper is written very well and explained in an organized way and the applications of machine learning with some physico chemical based model to identify efficient inhibitors against 3CL-protease.please address the following questions in the manuscript. I am recommending for the publication of this article after the following recommended changes in the manuscript:

1. In page 26, figure 2, from CMP5 until CMP10 they have all symbol (C) while all compounds into the figure has symbols from (a) to (j), Please correct it.

Response: Thank you for the suggestions. The Figure 2 has been modified accordingly in the revised manuscript.

2. In page 27, it is the same as figure 2 , all copunds into the figure has symbols from (a) to (j) while into the caption most of them has symbol (C)

Response: The Figure 3 has been modified accordingly in the revised manuscript.

3. Clarify what is the violet colors into the table 1

Response:Thanks for the suggestion. The description is added in the footer of the table.

---

## [Decision Letter · Decision Letter 1]

28 Mar 2023

In-silico Approaches for Identification of Compounds Inhibiting SARS-CoV-2 3CL Protease

PONE-D-23-03061R1

Dear Dr. Khan,

We’re pleased to inform you that your manuscript has been judged scientifically suitable for publication and will be formally accepted for publication once it meets all outstanding technical requirements.

Kind regards,

Ahmed A. Al-Karmalawy, Ph.D.

Academic Editor

PLOS ONE

Additional Editor Comments (optional):

Reviewers' comments:

Reviewer's Responses to Questions

**Comments to the Author**

1. If the authors have adequately addressed your comments raised in a previous round of review and you feel that this manuscript is now acceptable for publication, you may indicate that here to bypass the “Comments to the Author” section, enter your conflict of interest statement in the “Confidential to Editor” section, and submit your "Accept" recommendation.

Reviewer #1: All comments have been addressed

Reviewer #2: All comments have been addressed

Reviewer #3: All comments have been addressed

2. Is the manuscript technically sound, and do the data support the conclusions?

Reviewer #1: Yes

Reviewer #2: Yes

Reviewer #3: Yes

3. Has the statistical analysis been performed appropriately and rigorously? 

Reviewer #1: Yes

Reviewer #2: Yes

Reviewer #3: Yes

4. Have the authors made all data underlying the findings in their manuscript fully available?

Reviewer #1: Yes

Reviewer #2: Yes

Reviewer #3: Yes

5. Is the manuscript presented in an intelligible fashion and written in standard English?

Reviewer #1: Yes

Reviewer #2: Yes

Reviewer #3: Yes

6. Review Comments to the Author

Reviewer #1: The authors adequately responded to all comments and made the requested suggestions. The current revised form of the article can be accepted for publication

Reviewer #2: The revised manuscript is now suitable for publication after the authors made the necessary modifications.

Reviewer #3: The paper is explained on the right way and well implemented. Besides, most of all modifications have been carried out

7. PLOS authors have the option to publish the peer review history of their article (what does this mean?). If published, this will include your full peer review and any attached files.

Reviewer #1: No

Reviewer #2: No

Reviewer #3: **Yes: **Hazem Essam Okda

---

## [Editor Report · Acceptance letter]

5 Apr 2023

PONE-D-23-03061R1 

*In-silico* Approaches for Identification of Compounds Inhibiting SARS-CoV-2 3CL Protease 

Dear Dr. Khan:

I'm pleased to inform you that your manuscript has been deemed suitable for publication in PLOS ONE. Congratulations! Your manuscript is now with our production department. 

Kind regards, 

on behalf of

Dr. Ahmed A. Al-Karmalawy 

Academic Editor

PLOS ONE